# MACTA: A Multi-agent Reinforcement Learning Approach for Cache Timing Attacks and Detection

**Jiaxun Cui**[1]   **Xiaomeng Yang**[*2]   **Mulong Luo**[*3]   **Geunbae Lee**[*4]   **Peter Stone**[1,5]
**Hsien-Hsin S. Lee**[6]   **Benjamin Lee**[2,7]   **G. Edward Suh**[2,3]   **Wenjie Xiong**[†4]   **Yuandong Tian**[†2]

[1]The University of Texas at Austin   [2]Meta AI   [3]Cornell University   [4]Virginia Tech
[5]Sony AI   [6]Intel Corporation   [7]University of Pennsylvania
cuijiaxun@utexas.edu, yangxm@meta.com, ml2558@cornell.com, geunbae@vt.edu, pstone@cs.utexas.edu,
linear@acm.org, leebcc@seas.upenn.edu, edsuh@meta.com, wenjiex@vt.edu, yuandong@meta.com

## Abstract

Security vulnerabilities in computer systems raise serious concerns as computers process an unprecedented amount of private and sensitive data today. Cache-timing attacks (CTA) pose an important practical threat as they can effectively breach many protection mechanisms in today's systems. However, the current detection techniques for cache timing attacks heavily rely on heuristics and expert knowledge, which can lead to brittleness and the inability to adapt to new attacks. To mitigate the CTA threat, we propose using MACTA, a multi-agent reinforcement learning (MARL) approach that leverages population-based training to train both attackers and detectors. Following best practices, we develop a realistic simulated MARL environment, MA-AUTOCAT, which enables training and evaluation of cache-timing attackers and detectors. Our empirical results suggest that MACTA is an effective solution without any manual input from security experts. MACTA detectors can generalize to a heuristic attack not exposed in training with a 97.8% detection rate and reduce the attack bandwidth of RL-based attackers by 20% on average. In the meantime, MACTA attackers are qualitatively more effective than other attacks studied, and the average evasion rate of MACTA attackers against an unseen state-of-the-art detector can reach up to 99%. Furthermore, we found that agents equipped with a Transformer encoder can learn effective policies in situations when agents with multi-layer perceptron encoders do not in this environment, suggesting the potential of Transformer structures in CTA problems.

## 1 Introduction

With increasingly sensitive data and tasks, security in modern computer systems is recognized as one of the 14 grand challenges for engineering (National Academy of Engineering, 2007). As a concrete example, cache-timing attacks (CTA) in processor caches have been shown to leak private encryption keys (Yarom & Falkner, 2014; Liu et al., 2015), break existing security isolation (Kocher et al., 2019), cause privilege escalation (Lipp et al., 2018), and break new hardware security features in the latest processors (Ravichandran et al., 2022). In CTA, the attacker is able to gain such access to private information (*e.g.*, via memory access patterns) from the victim who shares a cache with the attacker. Over decades, the attack and defense policies in CTA have been explored manually by computer architecture experts. To defend against such attacks, statistical analysis and machine learning models with static strategies have been proposed for CTA detection, *e.g.*, CC-Hunter (Chen & Venkataramani, 2014) uses auto-correlation and Cyclone (Harris et al., 2019) uses an SVM classifier. Yet, new CTA attacks are still being reported (Xiong & Szefer, 2020; Briongos et al., 2020; Saileshwar et al., 2021; Guo et al., 2022b;a), showing higher leakage rates or the ability to bypass existing defensive mechanisms.

---

* Equal second-author contribution. † Equal supervision.
Code available at https://github.com/facebookresearch/macta.

Computer security can be seen as a competitive game between the attackers and the defenders, and game-theoretic approaches that analyze strategy (policies) for both sides have been proposed (Anwar et al., 2018; Elderman et al., 2017; Eghtesad et al., 2020). These methods highly abstract the attack and defense strategies, usually based on known attacks and defenses, and analyze simplified games in the limited strategy spaces. For example, Anwar et al. (2018) studies CTA-like attack scenarios where the attacker decides when to terminate its attack and the defender decides an abstract security level. However, real-world CTA has large action and state spaces for different agents, sparse reward, and long game horizons, making the game analysis hard without exploring all possible policies.

In this work, we use multi-agent reinforcement learning (MARL) to jointly explore and optimize complex attack/defense policies in CTA. We take an integrated approach of reinforcement learning and game theory. First, we build a multi-agent gym environment, **MA-AUTOCAT**, that closely models a realistic CTA setting and allows efficient learning for both attackers and defenders. Specifically, we study a detect-and-terminate defense. Second, we introduce and evaluate a MARL approach, named **MACTA**, to automatically find both attacker and detector policies through self-play, similar to past successes in games with large state/action spaces (*e.g.*, StarCraft (Vinyals et al., 2019), Go (Silver et al., 2016), and Poker (Brown & Sandholm, 2019)). MACTA adopts Fictitious Play (FP) (Brown, 1951), population-based training in MARL (Vinyals et al., 2019) and Proximal Policy Optimization (PPO) (Schulman et al., 2017) to learn the best response policy to a pool of diverse opponents, to avoid cyclic behaviors of the attacker/defender policies. Finally, MACTA uses a Transformer architecture to parameterize the policy/value function so that an important subset of actions can be picked up quickly during training, yielding fast policy learning.

We performed extensive experiments in a representative setting of cache-timing attack. The experiments show that learned policies trained with MACTA can *generalize* to detectors/attackers that they were not exposed to during the learning phase (henceforth referred to as "unseen detectors/attackers"). The MACTA detector exhibits a 97.8% detection rate on an existing human-designed attack without training on it and can lower the number of attacks per episode (bandwidth) of adaptive attackers by 20% on average. The MACTA attacker can bypass previously unseen detector, Cyclone, with a more than 99% success rate.

While there has been increasing interest and effort in using machine learning for computer system security recently, our work is the first to introduce the hardware timing attack problem as a promising application of MARL and show that MARL can be effectively applied to detect simulated CTA attacks with strong generalization.

Our main contributions are as follows:

- We contribute a simulated multi-agent environment MA-AUTOCAT that models realistic CTA and allows learning in both cache timing attacks and defenses.
- We introduce and evaluate MACTA, a multi-agent learning approach for CTA, and show the resultant detector acquires interesting high-level patterns that can generalize to novel attackers and make the cache less exploitable to high-bandwidth attacks.
- Our study on the neural architecture of learning agents indicates that the CTA is one case where Transformers are significantly better for retrieving state information than multi-layer perceptrons.

## 2   THE CACHE TIMING ATTACK CHALLENGE

The cache timing attack challenge is a fundamental problem to address as such kinds of attacks are stealthy but powerful. We leave the detailed reasons for studying the problem in Appendix A.1 and introduce the domain knowledge and problem formulation in this section.

### 2.1   DOMAIN DESCRIPTION

A cache is a small and fast on-chip memory commonly used in modern processor designs to reduce latency of memory accesses. Accessing memory addresses whose data are available in a cache is fast (called "*cache hit*"). If the data is not in the cache, data has to be retrieved from the main memory, which is much slower (called "*cache miss*").

Surprisingly, this timing difference in memory accesses due to caching could leak information across different programs/processes executing with a shared cache, a vulnerability known as *cache timing attacks* (CTA). As shown in Figure 1(a), CTA involves the attacker process and the victim process

both sharing the same cache. An example (Prime+Probe CTA (Liu et al., 2015)) is given in Figure 1(b). The victim's memory access will evict the attacker's cache line from the cache, causing latency changes in the attacker's future memory accesses. Thus, the attacker can infer whether the victim made access to a specific *memory address* by observing its own memory access latency, and thus be able to infer the victim's private information.

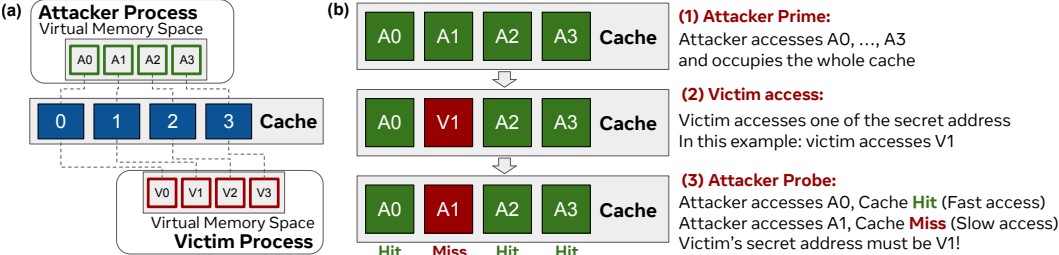

Figure 1: **(a)** Cache timing channel attack is formed when the attacker process and the victim process use the same locations of a shared cache for their memory accesses. **(b)** An example of Prime+Probe CTA in a 4-set direct-mapped cache. The attacker process can infer which memory address the victim process accesses by observing the latency.

## 2.2 PROBLEM STATEMENT

In this work, our goal is to jointly find novel attackers and robust detector policies that can generalize to unseen opponents, leading to insights for future cache design. The problem of joint learning can be formulated as a general-sum Partially Observable Markov Game (POMG), where the attacker and detector have limited observations and optimize for their own cumulative return. Given the finite set of policies, the resultant attacker is the best response to a mixture of all detector policies explored, and the resultant detector is the best response to a mixture of all attacker policies explored.

**Partially Observable Markov Games (POMGs)** Formally, an $n$-player episodic POMG can be described using a tuple $\{\mathcal{I}, \mathcal{T}, \mathcal{S}, \mathcal{P}, \{\mathcal{A}\}_{i=1}^n, \{\mathcal{O}\}_{i=1}^n, \{\mathcal{R}\}_{i=1}^n, \gamma\}$, where $\mathcal{I}$ is the finite set of players, $\mathcal{T}$ is the episode length, $\mathcal{S}$ is the true state space, $\mathcal{P}$ is the state transition probability. $\mathcal{A}_i$ is the action space of player $i$, and the joint action space of all agents is $\{\mathcal{A}\}_{i=1}^n = \mathcal{A}_1 \times \mathcal{A}_2 ... \times A_n$. Similarly, $\mathcal{O}_i$ is the observation space of player $i$, and $\mathcal{R}_i$ is the reward function for player $i$. Lastly, $\gamma \in [0, 1]$ is a reward discount factor. In POMGs, each agent only has access to its own observations and actions, and its goal is to maximize the cumulative episodic reward for itself given the opponents' policies, $\mathcal{J}^i(\pi^i, \pi^{-i}) = \mathbb{E}\left[\sum_t^T \gamma^t r_t^i | s_0, a_t^i \sim \pi^i(s_t), a_t^{-i} \sim \pi^{-i}(s_t)\right]$. A Nash Equilibrium (NE) is one solution concept to POMGs. Formally, a NE is defined as a saddle point where for any player's policy $\pi^i$, we have $\mathcal{J}^i(\pi_*^i, \pi_*^{-i}) \geq \mathcal{J}^i(\pi^i, \pi_*^{-i}), \forall i \in \mathcal{N}$. Namely, given all other agents' equilibrium policies $\pi_*^{-i}$, there is no motivation for agent $i$ to unilaterally deviate from its current policy $\pi_*^i$ to achieve higher returns.

## 3 MA-AUTOCAT

To study the learning dynamics of the attackers and the detectors in CTA, we develop MA-AUTOCAT, a gym (Brockman et al., 2016) environment that models realistic multi-agent CTA interactions. We build the environment based on a cache simulator, which faithfully models cache state changes, following practices in prior works on CTA detection schemes (Harris et al., 2019; Mirbagher-Ajorpaz et al., 2020). Note that experimenting detectors on real processors requires hardware modifications, which is prohibitively expensive. Figure 2 demonstrates the environment components and game mechanism.

In MA-AUTOCAT, each agent plays a different role, and each role has a specific goal (*i.e.*, reward), a different level of privileged accessibility (*i.e.*, observation) to the information of the environment, and a different way to take actions (*i.e.*, action space), listed as below:

***Benign Program (B)*** accesses memory in a regular way, implemented by replaying an offline log of memory accesses from regular programs (*e.g.*, a standard benchmark suite such as SPEC (Bucek et al., 2018)). It has no observation and no policy needs to be learned.

***Victim (V)*** accesses memory with addresses that depend on a secret. Studies have shown that such secret-dependent memory accesses are common in real-world applications (*e.g.*, HTTP parser), li-

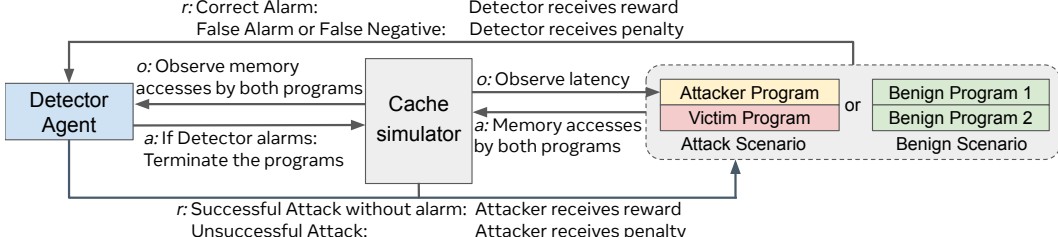

Figure 2: We propose MA-AUTOCAT, a multi-agent environment to jointly explore and optimize the policies of the attacker and the defender processes in CTA. In this environment, multiple agents can play different roles and learn from each other. The end goal is to learn policies that can generalize to deal with previously unseen opponents (e.g., those designed by human heuristics).

braries (*e.g.*, OpenSSL), and Linux kernel (Johannesmeyer et al., 2022; Qi et al., 2021; Oleksenko et al., 2020). In CTA, a victim's secrets usually contain multiple bits, and attackers target one bit at a time; after guessing one bit of a secret, the attacker moves to the next bit. To model this in our environment, the secret bit is reset after the attacker's attempt to guess the secret and the victim accesses an address depending on the secret when triggered.

***Attacker (A)*** aims to obtain the secret memory address of the victim process, by checking the patterns of latency of memory accesses. An attacker may learn a policy to *pick* which memory addresses to access, and observes the binary latency signal (slow/fast). The attacker can also *trigger* the victim process to execute, regain control after its execution, and *guess* the secret address of the victim if it is confident to do so. Importantly, the attacker can only see the latency of its own accesses.

***Detector (D)*** aims to raise the alarm as soon as possible when an attacker is present while avoiding a false alarm for benign programs. As a system process, we assume that the detector can observe memory accesses to the cache sets of all running processes in the environment. The detector will *terminate* an episode if an alarm is raised.

See Appendix A.2 for detailed specifications of the observations, actions, and rewards.

In each episode, we may pick multiple agents of different roles to be in the environment and let them interact. In this work, we mainly test the following two possible scenarios:

- Attack Scenario (**DAV**). The environment contains a detector, an attacker, and a victim. The attacker aims to obtain the secret address of the victim. The detector aims to detect the presence of an attacker and terminate processes as soon as possible.
- Benign Scenario (**DBB**). The environment contains a detector and two benign programs with no malicious intent. In this case, the detector should not raise any false alarms.

We leave more complicated settings, such as scenarios with both victims and benign programs (e.g., DAVB) as future work.

## 4 METHOD

The CTA that we consider is a POMG with three fundamental characteristics: **1) Partial Observability**. In CTA, the attacker knows which program to attack but can only see the attacker's own actions and latencies, while the detector does not know if there is any attacker nor which program the attacker is targeting. **2) Sparse-Reward Markov Game**. The CTA game can have a long episode length, and agents have to come up with a good action sequence before receiving the reward. Especially, the attacker must learn both low-level *skills* to perform attacks and high-level *strategies* to avoid defenders. **3) Environment Randomness**. Such randomness comes from randomized victim secret addresses and the random trajectory sampling of benign programs. We propose our method based on the three crucial features.

### 4.1 OUR APPROACH: MACTA

In this paper, we introduce our approach, MACTA (Figure 3 Right), as an initial solution to the CTA challenge using MARL. MACTA adopts Transformers (Vaswani et al., 2017b) as the neural encoder of policy nets, Proximal Policy Optimization (PPO) (Schulman et al., 2017) as the policy learning algorithm, and Fictitious Play (FP) (Brown, 1951) as the game-theoretic tool.

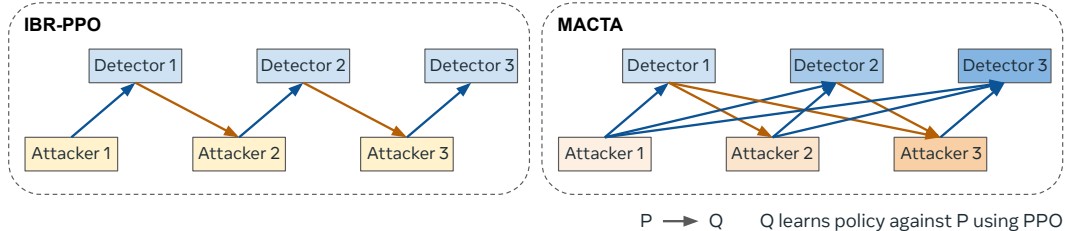

Figure 3: Method. Iterated Best Response PPO (IBR-PPO) learns the best response to the previous opponent only, while MACTA learns the best response to a uniform mixture of all historical opponents.

To deal with history-dependent partial observations and sparse rewards, both the attacker and the detector are equipped with policy nets with Transformer encoders. The Transformer encoder is mainly composed of scaled dot-product attention and multi-head self-attentions. It can effectively integrate information from long time horizons and large-scale data while not suffering from vanishing or exploding gradients in recurrent neural networks (RNNs) (Parisotto et al., 2020).

The attacker and the detector optimize their policies by the PPO algorithm to effectively learn a policy in the Markov game. Although independent reinforcement learning, where all agents are updating their policies simultaneously, is notoriously known for the instability issue in training (Tan, 1993), if we only train one agent at a time and keep others stationary, then other agents can be taken as a part of the environment, and PPO can effectively optimize the policy for higher cumulative rewards. Iterated Best Response PPO (IBR-PPO) (Figure 3 Left) is the most naive way of implementing the above idea. It alternates the training of the attacker and detector so that they learn the best policy against the most recent opponent. However, it may fall into the cyclic policy learning and never converge to any Nash Equilibrium (Roughgarden, 2010).

As a widely accepted method in MARL, creating a diverse pool of opponents and learning the best response to a mixture of them can alleviate the cyclic issue and help with generalization. Similar to fictitious play in game theory, we create a pool for each agent and add their historical policies to the pool. Concretely, for each iteration $\tau$, we denote the set of policies explored until $\tau$ of agent $i$ by our method as $\Pi_\tau^i$, the opponents' joint policy set as $\Pi_\tau^{-i}$. Then we learn the best response (BR), $\pi_*^i(\mathbb{U}(\Pi_\tau^{-i}))$, to the uniform mixture of the opponents' policy pool using a *best response learner* (*e.g.* PPO), and add the best response to the policy pool. Mathematically, for each iteration

$$\forall i : \Pi_{\tau+1}^i \leftarrow \Pi_\tau^i \cup \{\pi_*^i(\mathbb{U}(\Pi_\tau^{-i}))\}$$

where $-i$ represents all players except for player $i$, and $\mathbb{U}$ is the uniform distribution. There are more advanced meta game frameworks like Double Oracle (McMahan et al., 2003) and Policy Space Response Oracle (Lanctot et al., 2017), which measure the meta game payoff matrix among different explored policies and solves the matrix for the best opponent mixture. In our case, since the environment contains some randomness, it is inefficient to precisely estimate the payoff matrix. We thus leave exploring more advanced game frameworks as future work.

The above components constitute our approach (Algorithm 1 in Appendix A.7). MACTA alternates the training of attacker and detector every $E$ epochs and adds one *deterministic* policy checkpoint of the learning agent to the agent's policy pool every $N$ epochs. During one agent's training, the agent faces a *uniform* mixture of all opponents' past deterministic policy checkpoints. Note that we create such a mixture by uniformly sampling policies from the opponent's policy pool at each action step.

## 4.2 IMPLEMENTATION DETAILS

Specifically, we start with empty policy pools for both agents, first train the attacker for 50 epochs (each epoch contains 3000 training steps) to gain the basic skills of obtaining information from the victim program, and add one policy to the attacker's policy pool every 10 epochs. Then we stop the attacker's training and switch to train the detector against the pool of the first 5 attacker policies for 50 epochs. Similarly, the detector will have 5 policies by the end of this training iteration (50 epochs). The above process is repeated until the target training iterations (1800 epochs). We adopt an Actor-Critic implementation of PPO for both the attacker and the detector, and both the policy net and the value net are 1-layer 8-head Transformer encoders with different output heads. We leverage the RLMeta (Yang et al., 2022) learning framework for the PPO implementation, which is

an asynchronous version of PPO with sampling and learning in parallel, and construct our multi-agent learning framework on top of it. For stabilizing the self-play process, we also apply dual-clip PPO (Ye et al., 2020). Refer to Appendix A.7 for a more detailed description of training and environment hyper-parameters. Our code is available at https://github.com/facebookresearch/macta.

# 5 EXPERIMENTS

## 5.1 EVALUATION SETUP AND METRICS

To evaluate the proposed MARL method, we compare with a few attacker and detector baselines. For attackers, we consider a textbook attack Prime+Probe (Algorithm 2), an RL-based attacker (AutoCAT) (Luo et al., 2023), and the PPO with Iterated Best Response Oracle (IBR-PPO) attacker. For detectors, we include our implementation of CC-Hunter (Chen & Venkataramani, 2014) (Appendix A.8) and Cyclone (Harris et al., 2019) (Appendix A.8), and IBR-PPO Detector.

In this work, we employ episode return and intuitive metrics including ***Attack Correct Rate***, ***Attacks per Episode***, ***Detection Rate***, ***Episode Length***, and ***False Alarm Rate***. Details are listed in Table 1.

Table 1: Evaluation metrics.

| Metrics | Object | Description |
|---|---|---|
| **Attack Correct Rate** | Attacker | Measures the ability of an attacker to infer a secret correctly (attack successfully). It is the percentage of correct guesses among all guesses aggregated over episodes. |
| **Attacks per Episode** | Attacker, Detector | Measures the speed of an attacker or the attacker's ability to bypass detection or the detector's ability to prevent attacks. It is the average number of correct guesses per episode. |
| **Detection Rate** | Attacker, Detector | Detection rate is the percentage of DAV episodes alarmed by the detector within the time limit in the evaluated DAV episodes. |
| **Episode Length** | Attacker, Detector | Measures how fast the detector can find out the existence of the attacker. |
| **False Alarm Rate** | Detector | Measures the false positive (terminate episode before time limit) rate of a detector given all benign agents. |

## 5.2 BENIGN DATASET

We use the Standard Performance Evaluation Corporation (SPEC) 2017 benchmark suite (Bucek et al., 2018) to represent benign programs, and obtain their memory access traces using the gem5 simulator (Binkert et al., 2011). We then generate benign traces by combining the memory accesses from two programs based on the simulation timestamps. We introduce the details of the Train/Val/Test dataset in Appendix A.3.

## 5.3 RESULTS

All the experiment results below are reported on an 8set-1way L1 cache. The attacker's memory address range is 8-15 and the victim's secret address is randomly chosen between 0-7. The episode length is 64 steps. To evaluate different methods, we report the statistics based on three independent training instances for each learning-based method and control the final policies from different instances of a method undergoing the same number of optimization steps.

### 5.3.1 ATTACKER PERFORMANCE

Table 2: Attacker performance. Evaluation of the attacker's correct rate and number of attacks in 64-step episodes without detectors. Statistics are reported on three independent evaluations of 10,000 episodes.

| Attackers
Metrics | Prime+Probe | AutoCAT | IBR-PPO Attacker | MACTA Attacker |
|---|---|---|---|---|
| Attack Correct Rate (%) ↑ | $100.0 \pm 0.0$ | $100.0 \pm 0.1$ | $99.9 \pm 0.1$ | $100.0 \pm 0.1$ |
| Attacks per Episode ↑ | $3.0 \pm 0$ | $5.2 \pm 0.1$ | $5.2 \pm 0.1$ | $4.3 \pm 0.3$ |

We first evaluate the attacker agent's performance in terms of attack correct rate and the number of attacks in an episode, to validate that the attacker agent is conducting effective attacks. Table 2 shows that every attacker evaluated can achieve a decent attack correct rate, indicating the agent acquires

effective attack policies. In addition, the MACTA attacker has the smallest number of attacks per episode among the learning-based methods, because it learns to obfuscate itself as a benign program. Example attack sequences demonstrating the strategic attack behaviors can be found in Section A.4.

### 5.3.2 HEAD-TO-HEAD EVALUATIONS

In this head-to-head evaluation, we have an attacker play against a detector from *different* training instances for 10,000 episodes and report the mean detection rate and the mean episode length for all attacker and detector pairs. The head-to-head evaluation results can be found in Table 3 and Table 4. We also report the mean false alarm rate and the mean episode length of the detectors on *unseen* Benign agents in the last column of the table.

Table 3: Mean detection rate (%). Head-to-head evaluations with unseen opponents from different training instances. The higher the better for detectors when the opponent is an attacker, and the lower the better when the opponents are benign programs. '()' as Cyclone (SVM) is trained on Prime+Probe.

| Opponents / Detectors | Prime+Probe ↑ | AutoCAT ↑ | IBR-PPO Attacker ↑ | MACTA Attacker ↑ | Benign ↓ |
|---|---|---|---|---|---|
| CC-Hunter (thold=0.45) | $37.7 \pm 0.6$ | $13.7 \pm 1.3$ | $12.1 \pm 0.4$ | $16.4 \pm 2.3$ | $27.6 \pm 0.9$ |
| Cyclone (One-Class SVM) | $0.0 \pm 0.0$ | $55.8 \pm 4.3$ | $33.6 \pm 12.8$ | $9.0 \pm 5.3$ | $19.3 \pm 0.9$ |
| Cyclone (SVM) | $(99.5 \pm 0.1)$ | $0.0 \pm 0.0$ | $0.0 \pm 0.0$ | $0.1 \pm 0.1$ | $1.4 \pm 0.2$ |
| IBR-PPO Detector | $0.9 \pm 0.7$ | $7.3 \pm 20.5$ | $6.4 \pm 15.6$ | $8.4 \pm 21.9$ | $\mathbf{0.4} \pm 0.5$ |
| MACTA Detector | $\mathbf{97.8} \pm 0.9$ | $\mathbf{99.9} \pm 0.2$ | $\mathbf{99.6} \pm 0.4$ | $\mathbf{31.2} \pm 18.5$ | $1.1 \pm 0.2$ |

Table 4: Mean episode length (steps). Head-to-head evaluations with unseen opponents from different training instances. The lower the better for detectors when the opponent is an attacker, and the higher the better when the opponents are benign programs. Cyclone and CC-Hunter both require a fixed episode length of 64 steps.

| Opponents / Detectors | Prime+Probe ↓ | AutoCAT ↓ | IBR-PPO Attacker ↓ | MACTA Attacker ↓ | Benign ↑ |
|---|---|---|---|---|---|
| IBR-PPO Detector | $63.4 \pm 0.4$ | $59.6 \pm 12.4$ | $60.1 \pm 9.5$ | $58.9 \pm 12.2$ | $\mathbf{63.7} \pm 0.3$ |
| MACTA Detector | $\mathbf{16.4} \pm 1.1$ | $\mathbf{12.0} \pm 2.8$ | $\mathbf{12.5} \pm 2.2$ | $\mathbf{50.5} \pm 8.7$ | $63.4 \pm 0.1$ |

We find that the heuristic detector CC-Hunter cannot effectively discriminate the RL attackers from benign agents since the episodes are too noisy and too short to compute meaningful auto-correlations. Tuning the auto-correlation threshold only returns either a high false alarm rate or a low detection rate. The anomaly detector, Cyclone (One-Class SVM), is more effective at detecting high-bandwidth attackers such as AutoCAT and IBR-PPO attackers, yet it struggles with detecting low-bandwidth attackers like Prime+Probe and MACTA attacker and has a high false alarm rate. The SVM detector with Cyclone features is able to perform well (99.5% detection rate) on the heuristic attack (Prime+Probe) that it is trained on, but has low detection rate on RL attackers. Another drawback of these previous methods is that they require fixed-length observation that is longer than the steps needed to complete attacks (usually 12 steps in this cache configuration). IBR-PPO falls into the cyclic policy learning issue; the detector is able to react well (98.3% detection rate) to the attacker that it is trained against but fails to respond well to other attackers.

MACTA, however, is able to generalize to unseen attacks such as Prime+Probe and the IBR-PPO attacker. At the same time, MACTA also has a low false positive rate and fast detection speed which prevents further information leakage. We hypothesize that MACTA can abstract the general pattern of the attackers from interacting with diverse attacker strategies during training.

On the other hand, since the detector is trained to block all the previous attack policies, the attacker had to explore a new policy space to evade detection. The MACTA attackers are able to evade a variety of unseen detectors. The above findings highlight the benefits of using MARL solution concepts in learning the detectors.

### 5.3.3 EXPLOITABILITY EVALUATIONS

We measure how a detector can be exploited by adaptive attackers, by fixing a detector strategy and training an RL exploiter (*i.e.*, an attacker) against the detector by dual-clip PPO from scratch. The training curve of the exploiters of MACTA detectors can be found in Figure 4. As the training time of the MACTA detectors increases, it becomes more difficult for an RL attacker to bypass the detectors. Specifically, it will take the RL exploiter attacker longer to find a meaningful attack strategy. And even though the RL exploiter attacker can learn to attack eventually, the number of

attacks per episode decreases from around 5.0 attacks per episode (No Detector) to about 4.0 attacks per episode (MACTA-18th), leading to about 20% reduction in a learning attacker's bandwidth. The decrease in the number of attacks can come from the slower attack speed (to reduce the chance of detection) or faster detection speed so fewer attacks can be performed.

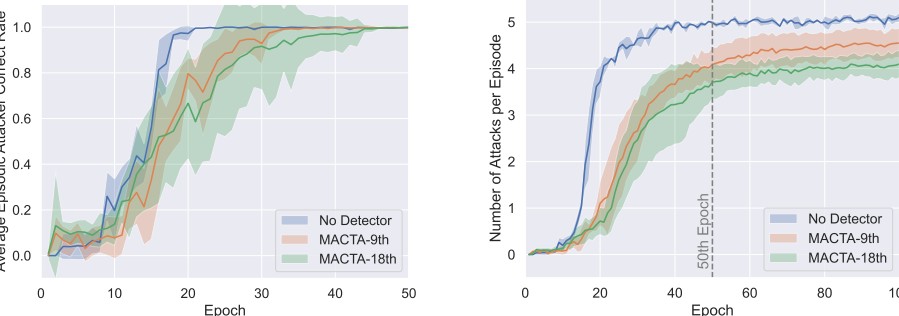

Figure 4: Exploitability evaluation. We fix the detector policies (No Detector, detector of 9th and 18th fictitious play iterations in MACTA (MACTA-9th, MACTA-18th)) and train an RL attacker against the detectors from scratch. **Left:** Average Episodic Attacker Correct Rate. **Right:** Attacker's Number of Attacks per episode.

## 5.4 NEURAL ARCHITECTURE STUDY

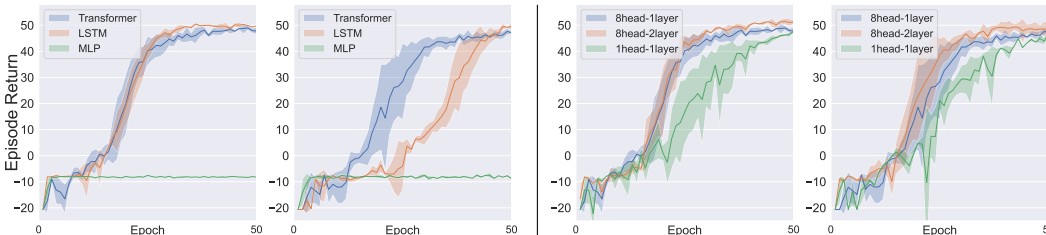

Figure 5: A study on neural architectures. We use a Transformer with 8-head attention and one Transformer encoder layer in MACTA experiments. **Left two:** Train attacker-only tasks using different neural architectures on two machines. **Right two:** Train attackers with different Transformer configurations on two machines.

Our CTA task is an example where neural architecture plays a critical role in learning a meaningful policy. We train attacker-only tasks using different network architectures on different machines (details in Appendix A.6) as shown in Figure 5. For PPO attackers, MLP with residual connections (He et al., 2016b;a) fails to achieve a high episode return, while the Transformer and LSTM (Hochreiter & Schmidhuber, 1997) networks succeed. For Transformers, our study shows that increasing the number of encoder layers in the Transformer can slightly improve the return but is less efficient in wall time. On the other hand, reducing the number of heads slows down learning. The above evidence suggests that the sequence modeling structure is critical for CTA attack policy learning. Our hypothesis is that a successful attack is composed of a series of events, which may contain history-dependent relations among events, and Transformers can effectively model such relations. While the prior work (Luo et al., 2023) also shows that Transformers can be used for RL CTA attacker, we provide more in-depth studies on different model architectures in this work.

## 6 RELATED WORK

**Detectors for Cache Timing Attacks** CC-Hunter (Chen & Venkataramani, 2014) proposes to detect cache-timing attacks using recurrent patterns generated during cache contention between attack and victim processes. More specifically, it uses autocorrelation to detect periodic interleaving between the two event trains. ReplayConfusion (Yan et al., 2016) records and deterministically replays a program's memory traces, changing the mapping of cache addresses but retaining the cadences. Executing the traces in different memory addresses can expose abnormal access patterns observed between an attacker and a victim, which do not exist in benign traces. Cyclone (Harris et al., 2019) uses cyclic interference from cache contention during an attack. This detector assigns domain tags to processes, then uses performance counters to enumerate abnormal cache contention behaviors triggered by each domain tag. PerSpectron (Mirbagher-Ajorpaz et al., 2020) trains a neural network classifier using the memory and latency event logs generated from attack examples. The follow-up work EVAX (Mirbagher-Ajorpaz et al., 2022) improves the classifier accuracy using generative

adversarial networks (GAN). Existing detectors based on known attacks cannot deal with evolving attackers. Our study shows that the RL attacker can learn novel strategies to bypass existing static detectors. MACTA solves this problem by enabling auto-discovery of attacker policies.

**Game Theory in Security Games**   Game theory provides a framework for decision-making and strategy, modeling how selfish agents interact and affect system outcomes. In Stackelberg games, a defender must first commit limited resources to protect disparate locations and an attacker that subsequently targets locations, potentially having seen the configuration of defenses (e.g., (Bier et al., 2007)). Such games have masked systems from probes (Schlenker et al., 2018), defended systems against varied attack types (Thakoor et al., 2020), and assigned human analysts to automated system alerts (Schlenker et al., 2017). Whereas Stackelberg requires the defender to move first, we consider how the defender's policy should respond to the attacker's evolving policy. Game theory inspires GAN for security (Zolbayar et al., 2021; Baimukan & Zhu, 2021). Unlike prior works that explore adversarial samples in the neighborhood of a heuristic attack policy, our RL approach explores a broader, unknown space of attack policies with a well-defined objective. RL is an instance of stochastic games, often modeled by a Markov Decision Process. Representative studies of such games for distributed systems include threat detection and resource allocation (Krishnamurthy et al., 2007; Fan et al., 2019). To the best of our knowledge, we are the first to formulate a stochastic game for realistic, practical hardware timing attacks.

**Population-based Multi-agent Reinforcement Learning**   Independent Reinforcement Learning in multi-agent environments suffers from the non-stationary opponent issue (Tan, 1993). While Iterated Best Response methods alleviate the above problem by learning from stationary opponents; they tend to over-fit to other players' policies and cause cycles in policy learning (Vinyals et al., 2019). Interacting with diverse opponent policies or heterogeneous agents is one effective way to avoid such cycles. Population-based MARL is thus proposed to solve large-scale extensive form games by creating a diverse pool of agents. Related work includes population-based reinforcement learning (Parker-Holder et al., 2020), Neural Fictitious Self-Play (Heinrich & Silver, 2016), Fictitious Co-Play (Strouse et al., 2021), prioritized self-play (Vinyals et al., 2019), Double Oracle (DO) (McMahan et al., 2003) and its generalization Policy Space Response Oracle (PSRO) (Lanctot et al., 2017). The most closely related applications of population-based MARL to security games, such as those of Eghtesad et al. (2020) and Wang et al. (2019), use variants of Double Oracle, but they deal with different and less stochastic domains than ours.

## 7    CONCLUSIONS AND FUTURE WORK

Our work explores the application of multi-agent reinforcement learning in the cache timing attack and detection domain. We first introduce the environment MA-AUTOCAT that allows learning for both attackers and detectors, and their complex interactions with caches. Then we propose to combine the game-theoretic concept of Fictitious Play and Proximal Policy Optimization to train both agents (MACTA). Empirically, we found that the detector generated by MACTA can capture the general pattern of attacks and generalize to unseen attacks. The exploitability study of the detector also indicates the detectors can impede the learning process of adaptive attackers and slow down the attacks. On the other hand, the MACTA attacker is able to explore new policy space and mimic the benign agents to bypass the detectors. Finally, the neural architecture study demonstrates the strong representation-ability of Transformers.

**Deployment in Real Systems**   We use a cache simulator to study CTA, but we believe the trained attacker and detector can be applied to real hardware with sufficient engineering efforts. For attackers, Luo et al. (2023) demonstrate that an attack pattern learned in a cache simulator can be applied to multiple Intel processors. Similarly, we also show that the attack sequences from a MACTA attack can work on commercial processors in Appendix A.5. For the detectors, with hardware changes, the neural network model can be deployed inside a processor with a reasonable area and power overhead, as demonstrated by Mirbagher-Ajorpaz et al. (2020).

**Convergence of the Policies**   In MACTA, we adopt the Transformer-based PPO algorithm as the policy learning oracle, so there is no guarantee that the algorithm will return the best response to the opponents in limited optimization steps. Meanwhile, little previous work discusses the convergence of Fictitious Play when it is used as a game-theoretic meta solver in the general-sum MARL setting. As training continues, we observe that the detector's ability to generalize slightly diminishes, indicating that it is forgetting some past attacks. We hypothesize it can relate to the convergence of one player's policy, which causes low policy diversity in the pool.

## ACKNOWLEDGEMENT

This work at UT Austin has partially taken place in the Learning Agents Research Group (LARG). LARG research is supported in part by NSF (CPS-1739964, IIS-1724157, FAIN-2019844), ONR (N00014-18-2243), ARO (W911NF-19-2-0333), DARPA, GM, Bosch, and UT Austin's Good Systems grand challenge. Peter Stone serves as the Executive Director of Sony AI America and receives financial compensation for this work. The terms of this arrangement have been reviewed and approved by the University of Texas at Austin in accordance with its policy on objectivity in research. Mulong Luo is partially supported by NSF grant ECCS- 1932501, and Geunbae Lee was partially supported by Commonwealth Cybersecurity Initiative. We thank Vishnu Kumar Kalidasan and John G. Harris at Virginia Tech for their technical support.

## STATEMENTS

**Ethical Statement**   Our work builds an environment enabling automated exploration of both attack and defense policies of CTA. The attacker agent may be used for developing new attacks maliciously. However, as shown in the vast amount of attack papers in computer security venues, exploring and understanding security attacks is a necessary and important step in developing future secure systems. Compared with prior works on security attacks on real systems Liu et al. (2015); Oren et al. (2015); Kocher et al. (2019); Lipp et al. (2018); Ravichandran et al. (2022), the MACTA agent only explores the attacks in a cache without considering other system-level activities. In addition, our framework results in a stronger detector, which can help design more secure systems.

**Reproducibility Statement**   Our experiments are reproducible in a reasonable range near the mean performance across multiple training instances reported. It is unavoidable that training with different random seeds or on machines with different hardware will result in different results, given the variance of explorations in reinforcement learning. However, our result is not a selection among multiple random seeds biased towards our benefits. We repeated the training for another three instances, giving us a similar result. We provide model checkpoints and publish the code at https://github.com/facebookresearch/macta.

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

# A APPENDIX

## A.1 WHY STUDY CACHE TIMING ATTACKS

CTA are stealthy but powerful. They do not violate any access control policies enforced by the operating system and low-level hardware and they are shown to pose serious security concerns in practice. For example, some implementations of security critical software such as encryption algorithms have a secret dependent access pattern, and an attacker can use CTA to obtain secret keys (Osvik et al., 2006; Liu et al., 2015). CTA also enables covert communication channels between two domains and breaks the existing security isolation mechanism, e.g., sandbox in javascript (Oren et al., 2015), isolation between processes (Kocher et al., 2019), and the system privilege levels (Lipp et al., 2018). CTA can also facilitate brute forcing hash values stealthily without triggering exceptions, which is shown to help break the ARM pointer protection mechanisms (Ravichandran et al., 2022). One of the important defensive strategies is to detect unique characteristics of memory access patterns of attacker programs that are different from usual benign ones, as leveraged by the state-of-the-art cache-timing channel detectors (Yan et al., 2016; Chen & Venkataramani, 2014; Harris et al., 2019; Mirbagher-Ajorpaz et al., 2020). However, many new attacks (Briongos et al., 2020; Luo et al., 2023) avoid the characteristics that the detector uses and it is hard to adapt existing detectors to previously unseen attacks or access patterns.

## A.2 ENVIRONMENT CONFIGURATIONS

Table 5: Environment hyper-parameters.

| Parameter Group | Parameter Name | Parameter Value |
|---|---|---|
| MA-AUTOCAT | Max Episode Length | 64 steps |
| MA-AUTOCAT | Observation Window Size for the attacker and the detector | 64 steps |
| MA-AUTOCAT | Probability between Attack Scenario and Benign Scenario during Training | [50%, 50%] |
| MA-AUTOCAT | Benign Program Logs (Train) | 48 Million Steps |
| MA-AUTOCAT | Benign Program Logs (Validation) | 4 Million Steps |
| MA-AUTOCAT | Benign Program Logs (Test) | 40 Million Steps |
| MA-AUTOCAT | Attacker Memory Address Range | 8-15 |
| MA-AUTOCAT | Victim Memory Address Range | 0-7 |
| Cache Simulator | Cache Configuration | L1 Cache, 8 set 1 way |
| Cache Simulator | Replacement Policy | Least Recently Used (LRU) |

**Game Mechanism.** In MA-AUTOCAT, within a fixed-length episode, the attacker agent can guess the secret address of the victim as many times as possible and get a reward for every correct guess (*successful attack*). In the meantime, the detector agent can monitor the cache access history and interactions of two programs and decide whether to raise a flag/alarm to terminate the episode to prevent further information leakage.

**Attacker's Reward Function.** The attacker is punished by 0.01 for every time step, +10 if guess the victim's secret successfully, -10 if incorrectly. It will get a one-time punishment of 20 if it reaches a timeout without any attack, a one-time punishment of -10 if identified by detector. The episode length to collect reward is affected by the detector.

**Detector's Reward Function.** The detector can raise a flag to terminate the episode. If the detector raises a flag in an attack scenario, then the detector receives a reward for remaining steps [max step - current step]; if the detector raises a flag in a benign scenario, then it receives a large penalty [$5 \times$ max step]. If the detector lets the episode going, for benign scenario there is no punishment; while the detector gets -10 every time the attacker attacks successfully.

**Attacker's Action.** For each time step, the attacker can choose an action $a^a \in \{a_X^a, a_v^a, a_{vr}^a, a_{gY}^a\}$, where $a_X^a$ represents access address X, $a_v^a$ represents letting the victim access a secret-related address, $a_{vr}^a$ represents letting the victim access a random address and $a_{gY}^a$ represents guessing the secret address to be $Y$.

**Detector's Action.** For each time step, the defender can choose $a^d \in \{a_{term}^d, a_{cont}^d\}$ where $a_{term}$ means terminate the episode and $a_{cont}^d$ means let the episode keep going.

**Attacker's Observation.** The attacker's observation space includes a history of attacker actions and memory access latency it receives from the cache simulator. For each time step, a new step observation $(s_{lat}^a, s_{vt}^a, s_{act}^a, s_{step}^a)$ is appended to the observation window, where $s_{lat}^a \in \{s_{hit}, s_{miss}, s_{N.A.}\}$ represents the access latency, $s_{vt}^a \in \{s_t, s_{nt}\}$ represents whether to wait for the victim's action, $s_{act}$ records the attacker's current action and $s_{step}^a$ is the current time step.

**Detector's Observation.** The detector can observe a history window of the memory access actions of both programs. For each time step, the new observation is composed of $(s_{lat}^d, s_{program}^d, s_{set}^d, s_{step}^d)$, where $s_{lat}^d \in \{s_{hit}, s_{miss}\}$ represents the access latency (access latency of all programs are visible to the detector), $s_{program}^d \in \{s_a, s_b\}$ indicates the identity of the program, $s_{set}^d$ represents the cache set being accessed at the current step, and $s_{step}^d$ represents the current defender time step.

**Benign Programs.** The benign programs share the same action space with the victim and the attackers, and their domain id is randomized per episode. The actions of benign programs are sampled from pre-collected memory traces of the benign programs and we map the traces to the cache configuration (8 sets and 1 way) in this paper. A detailed introduction to the benign datasets we use and the detectors' false alarm rate comparison is in Section A.3.

## A.3 BENIGN DATASET

The memory traces of different benign programs are sampled from the Standard Performance Evaluation Corporation (SPEC) 2017 benchmark suite (Bucek et al., 2018). Specifically, we run each individual benchmark in the gem5 simulator (Binkert et al., 2011) and generate log files containing all memory accesses executed and their corresponding timestamps. To sample typical memory access patterns, we follow the standard practice in computer architecture studies to skip the first 2 million operations which represent the warm-up phase of programs, and sample the memory accesses in the middle of execution by skipping more steps. We collect memory access traces from the following 10 SPEC benchmarks, of which names are in the form of (program_id.program_name_speed/rate): 500.perlbench_r, 502.gcc_r, 505.mcf_r, 548.exchange2_r, 549.fotonik3d_r, 602.gcc_s, 607.cactuBSSN_s, 631.deepsjeng_s, 638.imagick_s, and 641.leela_s. Since our benign scenario consists of two benign programs, we mix two memory traces based on the timestamp in the simulation using either different benchmarks or identical benchmarks with different starting points. We prepare each benign trace to have 4 million memory accesses in total. Note that there can be combinatorially many different traces of two programs, we only select a representative subset of them for the training and evaluation. Finally, we project the memory access traces onto the valid action space given the current cache configuration (8-set, 1-way, cache line size of 64 bytes).

We randomly select three programs (500.perlbench_r, 502.gcc_r, 505.mcf_r), and use different combinations (with replacement and with different skip steps) of them as training set. For example, the trace file name "500-2M_500-4M" means it contains two perlbench programs sharing the same cache. They start from different times (skipping the first 2 million memory accesses and skipping the first 4 million memory accesses, respectively) and continue until there are 4 million memory accesses by either program. We generate the validation dataset (549.fotonik3d_r, 607.cactuBSSN_s) and test dataset (548.exchange2_r, 631.deepsjeng_s, 638.imagick_s, and 641.leela_s) in the same way as the training set.

The ML models (MACTA, IBR-PPO, Cyclone) are trained on the same training set and tuned on the validation set. In the meantime, CC-Hunter's threshold is selected based on the validation set. Since the memory traces can exhibit different behaviors, we provide our per-trajectory false positive rate estimate in Figure 6. All the machine learning models appear to perform better on the test set than the training set, which indicates a potential distribution shift between the training set and the test set. CC-Hunter's false positive rate is too high (ranging from $7.5 - 30\%$) to be plotted in the figure. In general, MACTA and Cyclone detectors have similar false positive rate on benign programs. The IBRPPO detector has lower false positive rates but it also has much lower detection rates. In addition, we closely inspect the benign traces that cause false positive. We find most of them are variations of the Prime+Probe attack on subset of cache sets. This is because even though the benign programs do not have malicious intentions, but they can still generate small pieces of memory access patterns that happen to be an attack pattern.

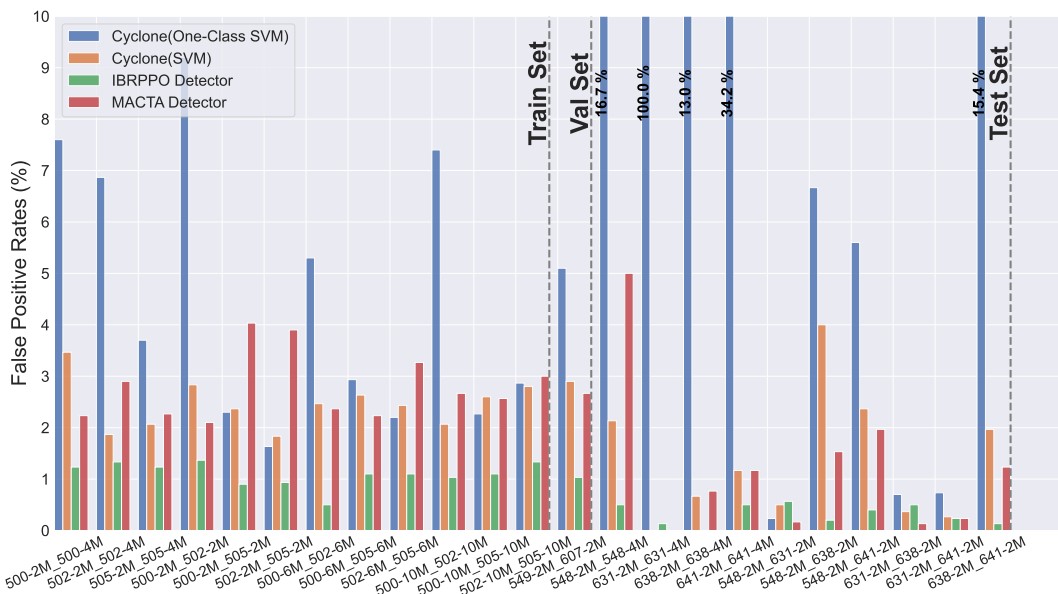

Figure 6: False positive rates on different datasets. We report the per-dataset mean false positive rate for three models. CC-Hunter(threshold=0.45)'s false positive rates are too high to be included here.

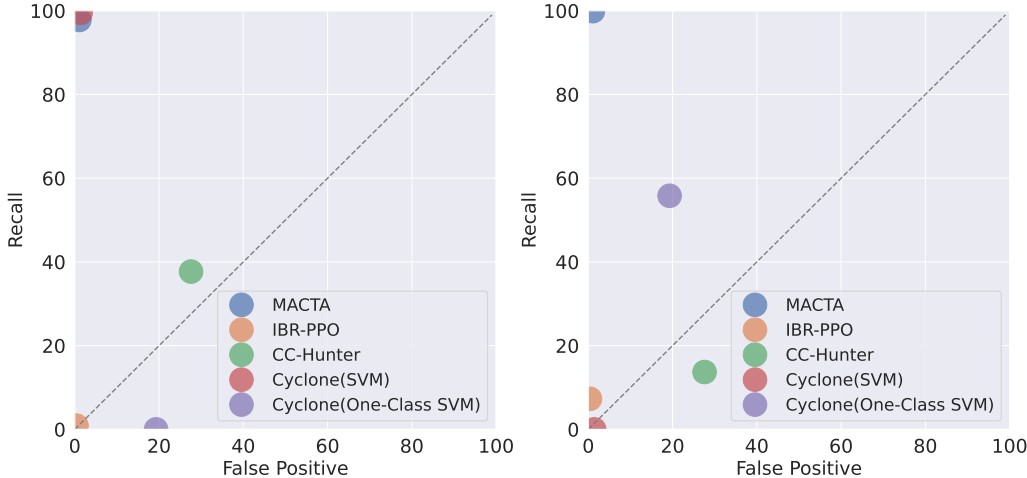

Figure 7: The relative positions of all detectors' performance on the ROC figure. The recall is shown for the Prime+Probe attacks (**Left**) and the AutoCAT attacks (**Right**). The false positive rate is measured on the proposed test benign dataset. Here, Cyclone is trained on Prime+Probe attack sequences. But we did not provide the Prime+Probe attack sequences to MACTA detector explicitly.

## A.4  TRAJECTORY ANALYSIS

Figure 8 illustrates different attackers' attack sequences given a fixed secret bit. Here, we use victim secret=5 as an example. The top row shows a sampled pattern of benign programs. In that case, the two programs act independently and alter the access to the cache frequently. The Prime+Probe attacker causes cache contention by accessing the cache frequently, and only invokes the victim to access the cache when needed. Once contention with the victim in one cache set is observed (i.e., a cache miss after the same address is accessed by the attacker), the attacker will make a guess without more memory accesses. The IBR-PPO attacker takes a similar strategy as AutoCAT's, but it learns to insert some extra victim invocation steps to confuse the detector. The issue with the extra invocation strategy is that the victim only accesses its secret bit, which can be easily captured by the detector. The MACTA attacker, however, learns more advanced strategies. It learns to invoke random victim

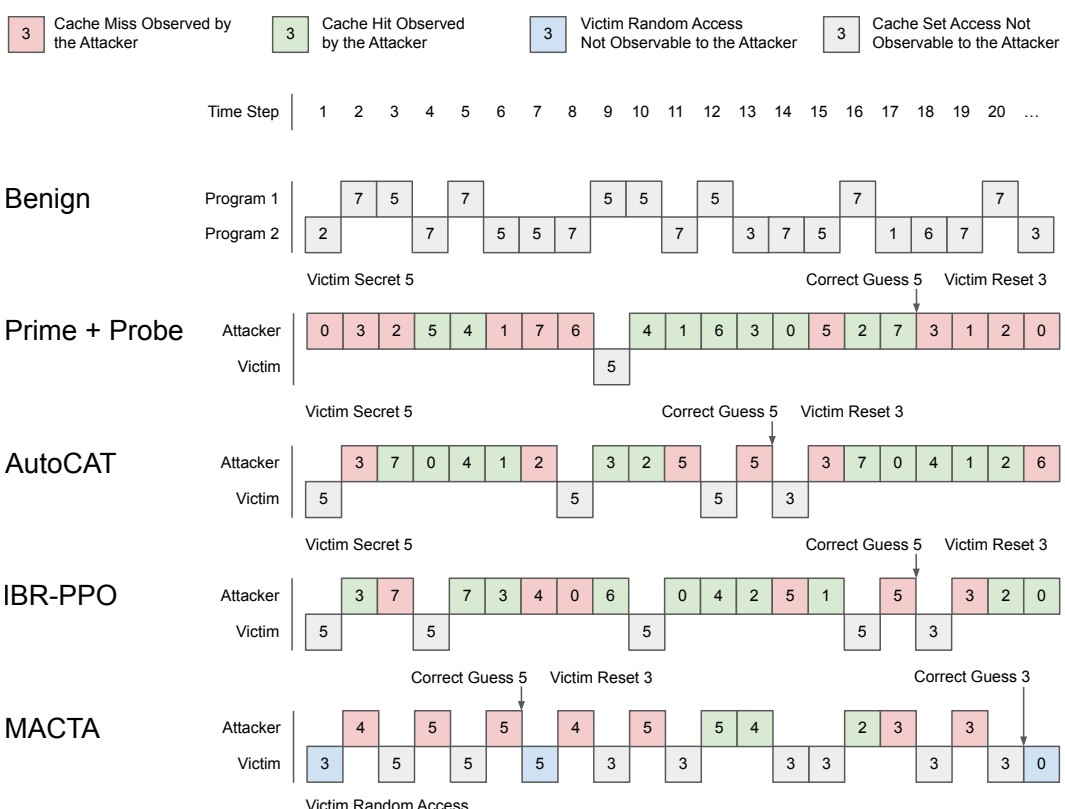

Figure 8: Example trajectories of different attackers and benign agents in a 8-set 1-way L1 cache. The number indicates the cache set being accessed. Red and green boxes show the observation by the attacker. The latency of other programs (i.e., victim or benign) cannot be observed by the attacker, but they can be observed by the detectors. The program IDs are randomized during training, and the attacker can be any of the two programs in the system. The cache is initialized with random states.

accesses to alter its behavior to be more similar to benign programs. Note that invoking random accesses from a victim can cause noise in the attacker's latency observation and make the steps needed for a successful attack longer. This means that to evade the MACTA detector, just inserting some extra victim invocation steps is far from enough. The attacker has to take a risk to invoke random accesses from a victim instead because the "easiest" policy space has been exhausted.

## A.5 REAL HARDWARE ANALYSIS

Table 6: Attack evaluation on commercial processors. We report the attack correct rates of MACTA attack sequences on three commercial Intel processors for 10,000 episodes. MACTA attackers achieve a $> 99.9\%$ correct rate in the simulator, and still $> 99\%$ on real hardware.

| CPU | Cache Level | #Sets | #Ways | Attack Correct Rate ↑ |
| --- | --- | --- | --- | --- |
| i7-6700 (SkyLake) | L3 | 8 | 1(partitioned) | 100.00% |
| i7-7700k (KabyLake) | L3 | 8 | 1(partitioned) | 99.97% |

We evaluated the effectiveness of the attack sequences produced by MACTA attackers on real hardware by running them on two commercial processors through CacheQuery (Vila et al., 2020). The attack sequences are generated from a MACTA attacker that is trained using a simulated environment for an 8-set 1-way cache configuration. Then, the attack sequences were performed on real hardware to obtain the attack correct rate. Table 6 demonstrates the attacker policy from the simulator can be transferred to real hardware with negligible discrepancy.

## A.6 MODEL ARCHITECTURES

In Section 5.4, we compared three different neural network architectures: Transformers (Vaswani et al., 2017a), LSTM (Hochreiter & Schmidhuber, 1997), and MLP. All of the methods are controlled at the same scale of parameters and trained with PPO (Schulman et al., 2017) without dual-clip (Ye et al., 2020) on two different machines. The details about the model architectures are listed below.

**Transformer.** In our experiments, all the policies use an 8-head 1-encoder-layer Transformer with $d_{model} = 128$ and $d_{feedforward} = 2048$. For the model architecture study, we study the changes in the number of heads in the multi-head attention mechanism, and the number of Transformer Encoder layers. Similar to Luo et al. (2023), we apply an average-pooling to reduce the step dimension.

**LSTM.** We employed a 1-layer LSTM with hidden dimension of 256. The input to the LSTM is the embedding of the pre-padding history of the observation. We concatenate the hidden and cell states of the last step and use it as the sequence embedding.

**MLP.** The MLP model we used directly feeds the input embeddings into 4 pre-activation residual blocks (He et al., 2016b;a) with hidden dimension 128. Each residual block is composed of 2 ReLU-Linear layers with a residual connection.

We obverse that the MLP model fails even when tested on multiple different machines while Transformer and LSTM models both work. The learning speed of Transformer and LSTM models can be different. Our hypothesis is that the CPU/GPU configurations of different machines may affect the policy lag (Petrenko et al., 2020) of Asynchronous PPO training, which may lead to different learning speed for different models.

## A.7 ALGORITHM AND TRAINING HYPER-PARAMETERS

The MACTA algorithm is explained in Algorithm 1, and detailed training hyperparameters can be found in Table 7. In MACTA, the policy pool is created by sampling a new model to do batch actions per step. It is an infrastructure implementation to produce faster sampling speed, so it is not strictly a per-step sampling, but it is per-step sampling considering a large amount of data. Additionally, we are exploring the per-trajectory policy sampling because it has nice theoretical properties. Nevertheless, it needs further infrastructure support.

---

**Algorithm 1** MACTA

---

1: Initialize Number of Fictitious Play Iterations $I$, Attacker Policy Pool $\mathcal{P}_A$, Detector Policy Pool $\mathcal{P}_D$, Number of Epoch per Fictitious Play Iteration $E$, Add a policy to Pool per $N$ epochs. $\mathrm{PPO}$ the Proximal Policy Optimization. $\mathbb{U}$ the uniform random sampling of policies per step. $i \leftarrow 0, j \leftarrow 0, k \leftarrow 0$
2: **while** $i < I$ **do**
3:     $j \leftarrow 0$
4:     **while** $j < E$ **do**
5:         $\pi_{Aj}$=$\mathrm{PPO}(\mathbb{U}(\mathcal{P}_D))$          ▷ Train attacker policy against the pool of the detectors
6:         **if** $j \bmod N - 1$==0 **then**
7:             $\mathcal{P}_A \leftarrow \mathcal{P}_A \cup \pi_{Aj}$              ▷ Add an attacker checkpoint to the attacker pool
8:         **end if**
9:         $j \leftarrow j + 1$
10:     **end while**
11:     $j \leftarrow 0$
12:     **while** $j < E$ **do**
13:         $\pi_{Dj}$=$\mathrm{PPO}(\mathbb{U}(\mathcal{P}_A))$          ▷ Train detector policy against the pool of the attackers
14:         **if** $j \bmod N - 1$==0 **then**
15:             $\mathcal{P}_D \leftarrow \mathcal{P}_D \cup \pi_{Dj}$              ▷ Add a detector checkpoint to the detector pool
16:         **end if**
17:         $j \leftarrow j + 1$
18:     **end while**
19: **end while**
20: **return** $\pi_A, \pi_D$              ▷ return the last policy of attacker and detector

---

Table 7: Training hyper-parameters for MACTA.

| Parameter Group | Parameter Name | Parameter Value |
|---|---|---|
| Fictitious Play | Fictitious Iterations | 18 iterations |
| Fictitious Play | Epochs per Iteration per Agent | 50 epochs |
| Fictitious Play | Training Steps per Epoch | 3000 steps |
| Fictitious Play | Frequency of Adding one Policy to Pool | 10 epochs |
| Computing Resource | Number of Sampling Actors | 72 Actors |
| Computing Resource | Sampling Instance per Worker | 3 Actors / Worker |
| Computing Resource | Remote Model Push Frequency | 10 steps |
| Computing Resource | GPU Information | 4 Nvidia Tesla V100 16G / 32G |
| Computing Resource | CPU Information | 80 Intel(R) Xeon(R) CPU E5-2698 v4 @ 2.20GHz |
| Proximal Policy Optimization | Replay Buffer Size | 262144 |
| Proximal Policy Optimization | Training Batch Size | 512 |
| Proximal Policy Optimization | Learning Rate | 1e-4 |
| Proximal Policy Optimization | Entropy Coefficient | 0.03 |
| Proximal Policy Optimization | Discount Factor $\gamma$ | 0.99 |
| Proximal Policy Optimization | Max Gradient Norm | 1.0 |
| Proximal Policy Optimization | GAE $\lambda$ | 0.95 |
| Proximal Policy Optimization | Policy Ratio Clipping $\epsilon$ | 0.2 |
| Proximal Policy Optimization | Value Clipping $\epsilon$ | 0.2 |
| Proximal Policy Optimization | Value Loss Coefficient | 0.5 |
| Proximal Policy Optimization | Dual-Clip Threshold | 3.0 |
| Model Architecture | Number of Transformer Encoder Layers | 1 |
| Model Architecture | Transformer d_model | 128 |
| Model Architecture | Transformer nhead | 8 |
| Model Architecture | Transformer dim_feedforward | 2048 |
| Model Architecture | Transformer dropout | 0.0 |

## A.8 HEURISTIC CACHE TIMING ATTACKS AND DETECTORS

### A.8.1 HEURISTIC ATTACKER ALGORITHMS

**Prime+Probe** (Algorithm 2) Osvik et al. (2006). First, in the prime phase, the attacker fills the cache set with its address value (lines 3) in a randomized way, then waits for the victim to access the cache set. Next, the victim accesses one of the cache sets, then replaces the loaded address value with its address (lines 5). Lastly, in the probe phase, the attacker accesses the cache sets again in a random permutation order, then measures the cache latency to load each set of the primed address value (lines 7 to 8). In a cache set accessed by the victim, the attacker observes increased latency (cache miss) and makes a guess.

---

**Algorithm 2** Prime+Probe Attack

---

1: $step \leftarrow step + 1$
2: **if** $step < len(attacker\_address\_range)$ **then**
3:     action = $prime\_address(step, cache\_size)$     ▷ attacker fills cache by attacker's address
4: **else if** $step = len(attacker\_address\_range)$ **then**
5:     action = $trigger\_victim(step)$     ▷ victim accesses a cache and fills its own address
6: **else**
7:     action = $probe\_address(step, cache\_size)$     ▷ attacker access caches again
8:     measure $latency(action)$
9: **end if**
10: **if** $latency = 1$ **then**     ▷ attacker observes for any cache miss
11:     action = $guess(action, cache\_size)$     ▷ attacker makes a guess on a victim's secret address
12: **end if**
13: **Return** action

---

A.8.2 DETECTOR ALGORITHMS

**CC-Hunter** Chen & Venkataramani (2014). Cache timing channels rely on the latency of events to perform timing modulation. To send information, two processes (*i.e.*, the trojan and the spy) generate a sufficient number of alternating conflict events (cache misses) to allow the adversary to decode the transmitted bit based on the average memory access times (hit/miss). Those behaviors show periodic, oscillating patterns of conflicts between the two processes. Therefore, autocorrelation is used to identify those patterns. Autocorrelation is the correlation coefficient of the signal with a time-lagged version of itself, along with measuring the event train $X$, as a variable at a time instance of $t$. Two cases of conflict miss, *i.e.*, either the victim eviting the attacker's cache line or the attacker evicting the victim's cache line, are considered for the event trains. For example, we can check the autocorrelation $C_p$ at a time lag $p$, which is expressed as:

$$C_p = \frac{\sum_{i=0}^{n-p} \left[ \left( X_i - \bar{X} \right) \left( X_{i+p} - \bar{X} \right) \right]}{\sum_{i=0}^{n} \left( X_i - \bar{X} \right)^2}$$

If there exists a time lag $p$ which $1 \le p \le P$, where $P$ is a predefined parameter such that makes $C_p$ larger than a threshold value, then it is assumed as an attack.

We tune the threshold to be $p = 0.45$ on our validation set, and this threshold yields a $7.5\%$ false alarm rate and a $38\%$ detection rate on Prime+Probe. However, this threshold fails to generalize to the test set, giving us a $27\%$ average false positive rate, as reported in this paper. The main issue with applying CC-Hunter to our environment is that our episode length is short, and the cache is initialized with random loads after resetting. As a result, even attackers' latency histories can have low auto-correlations.

**Cyclone** Harris et al. (2019). The concept of *cyclic* interference is commonly found in all known cache contention side-channel attacks and has been used for detecting those attack patterns. Interference occurs from the attacker to the victim process or vice versa, considered directional, and affects the behavior of microarchitecture in a disruptive manner. The cyclic interference can be noted as (a ⤳ b ⤳ a), where interference (a ⤳ b) is followed by (b ⤳ a). However, interference including a third party between attacker and victim, like (a ⤳ b ⤳ c), is not considered as cyclic interference. To distinguish attack and benign patterns, Cyclone uses a one-class support vector machine (One-Class SVM). In our experiments, however, we found that the one-class SVM is not effective in detecting our Prime+Probe implementation and has a high false positive rate under our testing configuration. Consequently, we also experimented with an SVM trained to classify Prime+Probe attack and benign traces, as the Cyclone paper suggests the feature can be used for other classifiers.

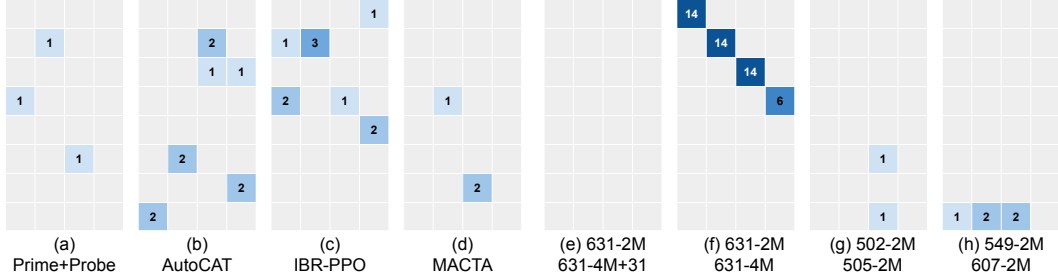

Figure 9: Example Cyclone features for various scenarios: (a, b, c, d) represent typical features when attackers interact with a victim; (e, f, g, h) depict typical features resulting from interactions between benign programs. The feature value in the grey areas is 0, and the intensity of the blue color indicates the frequency of cyclic inference, with darker shades representing more frequent occurrences. (e) illustrates the interaction between program 631.deepsjeng_s starting at 2 million (M) steps and the same program at 4M+31 steps. (f) demonstrates a trace of 631.deepsjeng_s self-mix from the test set, (g) shows a trace from the training set, and (h) presents a sample trace from the validation set. Typical test set features are similar to those of the train and validation set.

To extract the programs' cyclic features, we use 8 buckets and an observation window of 17 steps (which is the same as Prime+Probe's frequency). We train both Cyclone (One-Class SVM) and Cyclone (SVM) models with Gaussian kernels using 24,000 normalized train samples until convergence. For Cyclone (One-Class SVM), we choose the upper bound on the fraction of training errors and a lower bound of the fraction of support vectors to be 1%, with all training data generated from

the benign (DBB) scenario. For Cyclone (SVM), the regularization coefficient $C$ is set to 100, with 50% of the train data from benign scenarios, and 50% of them from malicious (DAV) scenario.

We observed that Cyclone (One-Class SVM) fails to detect Prime+Probe attacks and exhibits high false positive rates on datasets where two programs execute identical tasks. This inability to detect Prime+Probe is primarily due to the low bandwidth of the implemented attack. Regarding the elevated false positive rates, several factors contribute to this issue. One reason is that the cache configuration and system setting we employ could potentially bring high cyclic behaviors even between two benign programs. In addition, we discovered that when two benign programs with periodic memory access patterns are mixed, the difference in the starting time of the two programs could significantly impact the cyclic behaviors between the programs. Notably, for 631-2M_631-4M in the test set, Cyclone (One-Class SVM) exhibits a 100% false alarm rate (Figure 9(f)); however, if we shift the start time of the program, the false alarm rate decreases to 0% (Figure 9(e)). Cyclone (One-Class SVM) also demonstrates fluctuations with the program start time on the 641 self-mix but is generally more stable on other program combinations. In contrast, we found that the MACTA detector and Cyclone (SVM) are not sensitive to the program start-time offsets or combinations of programs, consistently showing low false alarm rates across various test sets and offsets. For fair comparisons, we report the statistics of all methods on the same test set without counterfactual dataset selection, but we encourage the readers to be mindful of the potential bias.

