# OpenReview forum: "MACTA: A Multi-agent Reinforcement Learning Approach for Cache Timing Attacks and Detection"
_ICLR.cc/2023/Conference — ICLR 2023 poster_

### Official Review · Reviewer_9uNr · 2022-10-16

**Confidence:** 2
**Correctness:** 3
**Technical Novelty And Significance:** 3
**Empirical Novelty And Significance:** 3
**Recommendation:** 6

**Clarity, Quality, Novelty And Reproducibility:**

Clarity & Quality: Problem statement of CTA is clear, but some details on RL does not address well.

Novelty: Considering CTA with MARL (game-theoretic) approach is an unprecedented view to tackle the issue.

**Strength And Weaknesses:**

Strength:

1. Converting detection of CTA problem into a multi-agent reinforcement learning is a fresh perspective to consider attack and detection in the area of data security and privacy.

2. This work builds CTA-related gym for multi agents to benefit detect-and-terminate defense strategy.

Weakness:

1. Since this work is based on game theory, it is good to provide a theoretical formalization on the problem and related proofs. Otherwise, it looks less sound and people may wonder the ability of this approach.

2. This work does not address explore-exploit tradeoff very well since it is usual in common RL scenario.

3. Although transformer is popular and good for many sequence learning tasks, this work does not explain enough on why transformer works well in the MARL for CTA. Authors may try to consider advantages of transformer for CTA in MARL from universal-approximation perspective like “ARE TRANSFORMERS UNIVERSAL APPROXIMATORS OF SEQUENCE-TO-SEQUENCE FUNCTIONS? ([https://openreview.net/pdf?id=ByxRM0Ntvr](https://openreview.net/pdf?id=ByxRM0Ntvr))” published on ICLR 2020.

**Summary Of The Paper:**

Cache-timing Attack (CTA) endangers privacy and security of data. However, traditional detection methods are not extendable. This work tackles CTA and detection from game-theoretic perspective by proposing a deep multi-agent reinforcement learning (MARL) approach with transformer for attackers and detectors.

**Summary Of The Review:**

Instead of heuristically iterating each possible CTA by human experts, this work provides a novel view from game-theoretic (to be specific, RL) perspective to address the issue. Compared to ML approach, this work does not simply learn behaviors of possible CTA, but it allows attackers and detectors to interactively develop accordingly. Although this work is lack of some theoretical explanations on MARL for CTA, the experiments themselves look good if they are not cherry-pick and it still worths to consider as an application work. I would like to hear from authors especially on W2 and W3, and I will also consider other reviews to make the final decision later.

---

> ### Author Response · Authors · 2022-11-11
> **Initial Response to Reviewer 3**
>
> Thank you for reviewing our paper!
>
> **Game theory formalization on the problem and related proofs**
>
> - If we make nice assumptions that the two-player game is fully observable,  zero-sum, and identical interest, then there is a work this year proving the convergence to Nash Equilibrium of a similar method which uses Q-learning as the best response oracle and Fictitious Play(FP) as a game theoretic oracle (Ours is PPO+FP): Fictitious Play and Best-Response Dynamics in Identical Interest and Zero-Sum Stochastic Games (Baudin et al., ICML2022)
> - The proof is not straightforward to extend because the problem we study is even more complex to analyze. It is a general-sum partially observable asymmetric Markov game, and we used a policy gradient method to approximate the best responses. So we provide empirical evidence in this work.
>
> **Explore-exploit tradeoff**
>
> - We handle the explore-exploit trade-off by enabling the entropy regularization in the Proximal Policy Optimization implementation. It encourages the learned attacker/detector to be as random as possible while maintaining policy optimality.
> - If a detector is too strong in the first place, it may be too hard for an attacker to find other meaningful attacks within the time limit (Our Figure 4 demonstrates this). Our framework allows the attacker to interact with a uniform mixture (population) of detectors, which is procedurally stronger so that the attackers can quickly pick up another best response policy. We experimented with different time limits for each self-play iteration and found that 50 epochs are enough for an agent to learn meaningful policies. It is to be studied if we should gradually allow longer learning time for each agent as the training goes on.
> - There are at least two types of possible results of the MARL training in our problem: (1) Type I: The detector ends up detecting a universal pattern of attackers, and there is no way for an attacker to fully evade the detection (2) Type II: The attacker looks exactly like the benign programs that it is impossible for a detector to distinguish. We are closer to the type I result, but we found it is still possible for an exploiter to exploit the detector if trained from scratch (Figure 5). So it is vital to keep the attacker policy diverse during the training if we are trying to find either type I or type II results.
>
> **Why the transformer works well in the MARL for CTA**
>
> - That’s a good question that we would like to investigate further. The inputs to the attacker and the detector are both memory access sequences (from different perspectives) with time stamps (positions) in CTA. So the input is permutation invariant, while MLPs need significantly more samples to model such permutations. The CTA attack is composed of a series of events that might have graphical/sequential inter-relations, which can be modeled by Transformers well (Yun et al., 2020) but not easily captured by MLP. Similar conditions apply to the detector as it observes action sequences of programs.
> - The RL community has explored the representation-ability of Transformers (Parisotto et al., 2020) in the memory environments or LSTM (Ni et al., 2022) in the partially observable environment and empirically showed the necessity of sequence modeling in RL in such cases.
>
> ---
> *Reference*
>
> [1] Parisotto, E., Song, F., Rae, J., Pascanu, R., Gulcehre, C., Jayakumar, S., Jaderberg, M., Kaufman, R.L., Clark, A., Noury, S. and Botvinick, M., 2020, November. Stabilizing transformers for reinforcement learning. In International conference on machine learning (pp. 7487-7498). PMLR.
>
> [2] Ni, T., Eysenbach, B. and Salakhutdinov, R., 2022, June. Recurrent model-free rl can be a strong baseline for many pomdps. In International Conference on Machine Learning (pp. 16691-16723). PMLR.
>
> [3] Baudin, L. and Laraki, R., 2022, June. Fictitious Play and Best-Response Dynamics in Identical Interest and Zero-Sum Stochastic Games. In International Conference on Machine Learning (pp. 1664-1690). PMLR.
>
> [4] Yun, C., Bhojanapalli, S., Rawat, A.S., Reddi, S. and Kumar, S., 2019, September. Are Transformers universal approximators of sequence-to-sequence functions?. In International Conference on Learning Representations.

---

### Official Review · Reviewer_Mrpg · 2022-10-21

**Confidence:** 4
**Correctness:** 3
**Technical Novelty And Significance:** 2
**Empirical Novelty And Significance:** 2
**Recommendation:** 6

**Clarity, Quality, Novelty And Reproducibility:**

Overall, the presentation of the paper is satisfactory.

The quality of the English text is mediocre.

Figures and Tables are mediocre (there are some inaccuracies)

The topic addressed by the manuscript is niche, but not very investigated and relevant for ICLR.

The references are appropriate.

The contribution is (potentially) significant.

The novelty is not high: ultimately, the paper takes a well-known approach (MARL) and applies to solve a security problem. However, doing so is not new, and the results are expected.

The reproducibility is “fair”: the authors “promise” to release the code, and the appendix is filled with details. However, from my own experience, reproducing the testbed can be very hard without disclosing the code. So I trust the authors to truly do so in case of paper’s acceptance.


**Strength And Weaknesses:**


STRENGTHS:
+ Simple, but interesting methodology
+ Easy to understand
+ The experiments consider several baselines


WEAKNESSES
- Poor novelty
- Lack of a “realistic” use case
- Misleading advantage
- Everything is simulated


**Summary Of The Paper:**

(before I begin, I report that he text of the paper contains hyperlinks to external URLs (e.g., “14 grand challenges of engineering”). I invite the authors to remove these and report the links either as footnotes or actual references, as it is not possible to appreciate them by those people that read the paper on printed-paper (aside from representing a security risk, as the URL is hidden). )


The paper tackles the problem of cache timing attacks (CTA) detection. Specifically, the paper proposes MACTA, a method that leverages “Multi Agent Reinforcement Learning” (MARL) to simultaneously launch CTA, and detect them. Experiments on a synthetic testbed show that the attacks “learned” by the “attacking agent” developed through MACTA can bypass existing detection methods; whereas the “detection agent” developed through MACTA can not only detect the attacks of the “attacking agent”, but also (to some degree) those tailored for detection methods employing handcrafted heuristics, or traditional supervised learning techniques.


**Summary Of The Review:**

I liked reading the paper, and it was very hard to find a “recommendation”. On the good side, the paper tackles an underinvestigated problem (at least in the context of Machine Learning for Cybersecurity), and the experimental evaluation is interesting enough to (potentially) inspire future work in this domain. On the bad side, the paper falls short in providing a significant advancement in the context of machine learning, some results are underwhelming/expected, and the testbed is entirely simulated.

Put simply, I think this paper would be **perfect** as a workshop paper, but it is not of good enough quality for ICLR. I recommend a “borderline reject”. I will provide below an explanation of all the major issues that affect the paper, alongside a set of questions and potential improvements. I look forward to reading an “enhanced” version of this paper. Furthermore, I also stress that I will gladly engage in a discussion with the authors.


**Unfair novelty claim.**
According to the introduction, “our work is the first to show a real-world hardware security problem can be addressed by jointly improving both attacks and detection.” This claim is unfair: using reinforcement learning to both “attack” and “defend” a security system is not novel at all. For instance, Apruzzese et al. [A] (published in 2020) do exactly this. The fact that this paper shows this for a “hardware” security problem is not very relevant to support a novelty claim. (especially because the experimental evaluation is done on a simulation, and not on “real” hardware)

**Lack of a true “realistic” use case.**
According to Section 3, the paper will ultimately consider two cases: one in which the environment includes a malicious entity (in which the goal is *detecting the malicious actions*); and one in which the environment includes only benign programs (in which the goal is *avoiding the generation of false positives*). The paper clearly states that “We leave more complicated settings, such as scenarios with both victims and benign programs as future work.” While I appreciate the transparency, I do not think that such “omission” can be simply overlooked: a common issue in *real* detection settings is when the environment includes both benign and malicious activities, wherein the latter are typically a “needle in a haystack”. I acknowledge that these may be “hard” to model, but such a lack is a significant limitation of this paper – especially given that it is submitted to a premier venue such as ICLR. For this reason, the real-world impact of the proposed method is difficult to gauge.

**Misleading (claimed) advantage.**
The proposed method, MACTA, ultimately attempts to overcome the limitations of “human-defined” heuristics (quoting from the abstract: “However, the current detection of cache timing attacks relies heavily on heuristics and expert knowledge, which can lead to brittleness and inability to adapt to new
attacks.”). However, since the entire RL environment (i.e., action space, reward, observation) is designed also by humans, wouldn’t this simply move to problem to a different “setting”? In other words: the RL environment is *also* designed by humans by using their expert knowledge. At the same time, an “improper” generation of the environment can lead, to an agent that can (synthetically) learn devastating attacks (and to a detector that can detect such attacks) but which may be not realistically conceivable. In turn, this can also lead to blind spots, i.e., if the environment does not take into account all the factors that can be realistically modeled (this is an implicit limitation of RL approaches).
Put simply, I think that the authors should tone down the way MACTA is presented.

**Questions/issues on Abstract, Introduction and Section 2.**

•	The abstract is filled with subjective and vague terms to present the improvement of the proposed method. For instance, “substantially outperform” should be quantified in an objective way; “more stealthily” should also be numerically quantified. Finally, “less exploitable” is also unclear.

•	In both the Abstract and Introduction, the text reports a very vague statement: “The experiments show that learned policies trained with MACTA can generalize to unseen detector/attackers.” Specifically, what does the term “unseen” stand for? Does it mean attacks that have not been “seen” before the operations performed by the RL agent? In which case, it is “obvious” that the learned policy can also detect such attacks; conversely, if the policy can detect attacks that are not seen “even during the learning phase of the agent”, then the result would be slightly more impressive (note, however, that [A] also showed a similar finding).

•	Why is the environment called “MA-AutoCAT”? Shouldn’t it be “MA-AutoCTA”?

•	I think Section 2.1 is redundant. It presents information that is ultimately not necessary to understand the contribution of the paper, while at the same time not providing enough details to fully allow a non-expert reader to understand the issues of CTA. I invite the authors to remove such Section (perhaps putting it in an Appendix, and maybe expanding it). The current Section 6 can replace Section 2.1, and the space obtained by such “cut” can be used to expand on the technical contribution.

•	On this note, something I’m curious about is whether anything has been done to counter CTA not with RL, but with GANs – given the many overlaps between these two techniques.

**Questions/issues with Sections 3 and 4 (method):**

•	Section 3 “In our environment, the secret is reset after the attacker’s attempt to guess the secret and the victim will access an address depending on the secret when triggered.” Why? Please provide an explanation for this choice. If it is an original idea, then a description and a motivation is necessary. If it is drawn from well-known practices, then such practices should be referenced.

•	Figure 1 (caption): “Cache timing channel attack is formed when the attacker process and the victim process map their memory to the same system cache.” Shouldn’t it be “same address of the system cache”? Clearly, the attacker and the victim “must” use the same cache for a CTA to be conceivable…

•	Section 3: “Detector (D) aims to raise the alarm as soon as possible when an attacker presents while avoiding a false alarm for benign programs.” Presents should be “is present”?

•	Section 4: “The CTA is a POMG with three fundamental characteristics:” this is unclear. Does it mean that “any CTA is a POMG”, or that “the CTA we consider is a POMG”?

•	Section 4: “In CTA, the attacker knows whom to attack but can only see its own actions and latencies”. It is unclear whether “its” refers to the attacker, or to the target (“whom”).

•	Section 4: “Especially the attacker must learn both low-level skills to perform attacks and high-level strategies to avoid adversaries” are the “adversaries” represented by the “detector”? In general, avoid using “attacker” and “adversaries” with different meanings in the same sentence.

**Questions/issues on Experiments**

•	Section 5.2: the experiments are carried out by simulating an 8set-1way L1 cache. Modern CPUs have far “larger” caches. Why did the authors consider such a simplistic setting? I believe the overall feasibility of the approach has deep connections to the architecture of the cache, and choosing such a simplistic one may (unfairly) induce favorable results for MACTA over prior baselines.

•	Why is it that, in Table 3, the “IBR-PPO Detector” has 0 detection rate against ALL attacks (including the IBR-PPO attack)? Furthermore, it is unfair to “remove” the perfect detection rate of Cyclone from Table 3. Moreover, also concerning Cyclone, I do not agree with the statement that “fails to detect RL attackers”: its detection rate is 20%, which is only 5% less than the proposed MACTA; at the same time, Cyclone exhibits a much better false positive rate than MACTA (4.2 vs 11.3): in these cases, Cyclone is much better than what the authors describe.

•	In Section 5.2.1, the paper states that “MACTA has low detection speed”: where is this reported? Note that I do not refer to “epochs” but to actual “time”. Furthermore, what the paper does not report (afaik) is the amount of computational resources required to develop (i.e., write code and train) MACTA: indeed, I am inclined to believe that implementing MACTA requires a huge effort – potentially superior than the one used to develop, e.g, Cyclone.


EXTERNAL REFERENCES

[A]: Apruzzese, Giovanni, et al. "Deep reinforcement adversarial learning against botnet evasion attacks." IEEE Transactions on Network and Service Management 17.4 (2020): 1975-1987.

---

> ### Author Response · Authors · 2022-11-11
> **Initial response to reviewer 2 (part I)**
>
> Thank you for the great effort in reviewing our paper!
>
> **Unfair novelty claim**
>
> Please check the general response above, and we provide more details here.
> In addition to the much larger attacker’s action space compared to prior works, our evaluation of the generalization of detectors is also novel. We held out heuristic attackers (Prime & Probe) during training, and tested the detectors with such attackers to measure the detectors’ generalizability. Table 3&4 also demonstrates such generalizability by a tournament evaluation among attackers and detectors obtained from **different methods** or **same methods, different training instances**. That’s why (1) we do not report the cyclone’s detection rate on prime & probe in the Table3, and (2) the IBR-PPO detector can have 0 detection rate everywhere, because we do not test it with the attacker it has seen during training (We illustrated in the paper that this detector has a good performance 98.3% on the trained attacker).
>
> **Lack of realistic use case**
>
> Please check the general response above, and we provide more details here.
> - To clarify, “victim” refers to a benign program being attacked. In that sense, our evaluation always includes multiple programs, at least one benign.
> - On real hardware, when simultaneous multi-threading (SMT) is enabled, in each CPU core there will be at most 2 programs running simultaneously, sharing one L1/L2 cache. Thus, we focus on the attacker+victim vs. benign+benign setting in one core.
>   - For private L1/L2 caches, our setup is realistic because there can only be a small number of  programs (one or two run at a time) sharing the cache in one core of a hyperthreading processor;
>   - For L3 cache which is shared by multiple programs, our detectors can be applied pairwise to the running programs or applied to each program individually by treating one program as a potential attacker and all other programs together as a victim/benign program.
>   - Previous works such Cyclone (Harris et al., 2019) and CC-Hunter (Chen & Venkataramani, 2014) have also evaluated similar cases.
>
> **Everything is simulated**
>
> Please check the general response above, and we provide more details here.
> - Our claim of real-world problems contrasts the toy or game domain; we study an underlying security problem that has a broad impact.
> - Just like prior work on CTA detection(CC-hunter(Chen & Venkataramani, 2014),  Cyclone (Harris et al., 2019), PerSpectron (Mirbagher-Ajorpaz et al., 2020) ), our detector is meant to be implemented in hardware. Because off-the-shelf processors do not have necessary hardware, we use simulators to study the multi-agent scenario. In terms of cache behavior (including memory access and cache hit or miss), there is a negligible gap between a cache simulator and the real hardware. The main gap between simulation and real hardware is the actual timing, area, power, which is not in the observation space in this problem. The prior work also evaluates the detector in hardware simulators. We note that it is a common practice to study a new hardware design in the simulator.
> - The configuration studied is representative in CTA and is not biased toward the success of the MACTA.
> Evidence shows that generated attack sequences in the simulator can be applied to real hardware (Luo et al., 2022).
>
> **Misleading (claimed) advantage**
>
> - Previous detectors in the CTA heavily relied on derived features of attackers based on human observation and insights. For example, CC-Hunter is simply based on the high auto-correlation of the attacker trajectory and Cyclone selects features such as cyclic inference by observing common patterns of past attackers. But a learning attacker can evade detection based on above features by modifying its behaviors, as shown in previous work (Luo et al., 2022). In contrast, the input to our detector is close to the raw cache access sequence, so that allows the detector to derive invariant/common features of attackers automatically by exhausting the attacker’s policy space.
> - On the attacker side, new human-designed attackers that claim nice evasion (Xiong & Szefer, 2020; Briongos et al., 2020; Saileshwar et al., 2021; Guo et al., 2022b;a) are designed based on the known detectors. Our work automates the evolution of both attackers and detectors that are based on maximum available information, so that the generated attackers/detectors during training can explore in a less compromised or uncompromised strategy space.
> - In terms of the environment design, we feed all available information to both agents according to the threat model. We did not select a subset of information to be the observation space. The rewards are designed using the common metrics in communication, i.e., error rate (attack correct rate), bandwidth (attacks per episode), detection rate, etc. Although it is still human heuristic, such a design does not require a deep understanding of how the timing attack works, just the threat model of the attack.

---

> > ### Comment · Reviewer_Mrpg · 2022-11-12
> > **Ack**
> >
> > I thank the authors for their response.
> >
> > A few comments:
> >
> > ```
> > We have optimized our code to tune down the false positive rate of MACTA detector to be at the similar level as cyclone and the results will be reflected in the next revision of the paper.
> > ```
> >
> > I'm really interested in seeing these new results, and discuss the quality of MACTA in light of the performance shown by this comparison.
> >
> > ```
> > Just like prior work on CTA detection(CC-hunter(Chen & Venkataramani, 2014), Cyclone (Harris et al., 2019), PerSpectron (Mirbagher-Ajorpaz et al., 2020) ), our detector is meant to be implemented in hardware. Because off-the-shelf processors do not have necessary hardware, we use simulators to study the multi-agent scenario. In terms of cache behavior (including memory access and cache hit or miss), there is a negligible gap between a cache simulator and the real hardware. The main gap between simulation and real hardware is the actual timing, area, power, which is not in the observation space in this problem. The prior work also evaluates the detector in hardware simulators. We note that it is a common practice to study a new hardware design in the simulator.
> > ```
> >
> > In my review, I was not questioning the fact that MACTA "tackles a real-world problem", but rather that **the paper** makes abundant use of the term "real-world", but does not perform the evaluation "in the real-world". In this case, it is irrelevant what prior work has done, and my stance is that the paper is overclaiming (or rather, hinting at something that is not "proven").
> >
> > An evaluation on real-hardware would dramatically improve the contribution of this paper to the state-of-the-art. Without such an evaluation, I invite the authors to remove the emphasis on the "real-world": ultimately, any attack "can work in the real-world". The text can very well state that, e.g., "we tackle a hardware security problem. By following the best practices, we simulate a realistic testbed and show that MACTA is an effective solution. Our results suggest that MACTA can mitigate the threat of CTA, which can impact real-world hardware." (or something along these lines).
> >
> > I must confess that my original recommendation was probably influenced by the fact that I was hoping to see experiments on real hardware, and was left with just simulations (even if such simulations are sensible, I must stress that the paper was submitted to ICLR, hence my "bar" is set very high).
> >
> > ```
> > On the attacker side, new human-designed attackers that claim nice evasion (Xiong & Szefer, 2020; Briongos et al., 2020; Saileshwar et al., 2021; Guo et al., 2022b;a) are designed based on the known detectors. Our work automates the evolution of both attackers and detectors that are based on maximum available information, so that the generated attackers/detectors during training can explore in a less compromised or uncompromised strategy space.
> > ```
> >
> > Assuming an attacker that "knows" the detector is a common practice in security$^1$, since the intent is providing a defense that works against the "strongest" possible attacker; many reputable works$^2$ assume an attacker with "perfect knowledge". I am curious to know the authors' opinion on this matter.
> >
> >
> > $^1$ Arp, Daniel, et al. "Dos and don’ts of machine learning in computer security." Proc. of the USENIX Security Symposium. 2022.
> >
> > $^2$ Shan, Shawn, et al. "Gotta catch'em all: Using honeypots to catch adversarial attacks on neural networks." Proceedings of the 2020 ACM SIGSAC Conference on Computer and Communications Security. 2020.

---

> > > ### Comment · Reviewer_Mrpg · 2022-11-18
> > > **Revision?**
> > >
> > > Dear authors, I was wondering whether you were going to upload a ```next revision of the paper```. As far as I can tell, the uploaded version on OpenReview is still the original one.
> > >
> > > Given that the rebuttal period ends soon, I'd like to know if any such revision is going to be expected before the end of the rebuttal phase. I am willing to increase my score if the results are promising.

---

> > > > ### Author Response · Authors · 2022-11-18
> > > > **Revision will be uploaded before end of rebuttal**
> > > >
> > > > Dear reviewer, Thanks for the active involvement for the improvement of the paper! We are working on finalizing the revision of the paper and the response to your previous questions; The revision will be uploaded with a short summary of updates before the end of the current rebuttal phase.

---

> > > ### Author Response · Authors · 2022-11-19
> > > **Assumption on attacker knowing the detector**
> > >
> > > Dear Reviewer,
> > >
> > > We have uploaded the paper revision regarding (1) lowering the false alarm rate and (2) real hardware statement.
> > >
> > > Regarding the "perfect knowledge" assumption in the security community, here are our thoughts:
> > >
> > > - We agree that it is vital to ask the question “Is there a detector such that even if it is known by the attackers, there is no attacker that can evade it?”. We think the practices that assume an attacker with perfect knowledge about the detector can help evaluate the white-box robustness of a detector.
> > > - The security community essentially resembles a multi-agent learning environment by human experts: In research papers, once a new attack is proposed, it should demonstrate the capability of bypassing existing mitigations; conversely, a proposed detector needs to show its coverage of the past strongest attacker. Nevertheless, suppose a detector is only designed to detect the strongest attackers. In that case, chances are that it can not respond to "exploiter" attacks, as the attacker might fall outside the detector's capability. We agree that a certain static attacker and detector could be mitigated/broken by the next generation of detector and attacker. However, during such iterations, the security researchers build a system with stronger security guarantees over time. That is why we use the multi-agent method to help accelerate the iteration automatically with less manual effort.
> > > - As mentioned by (Shan, Shawn, et al., 2020) in section 7.2, their developed defender is still vulnerable to the exploiter attacker that "knows" the detector. Arp et al. (2022) point out that the machine learning defensive system suffers from adaptive adversaries that specifically targets it, especially when it relies on only a few features. We take a dynamic view of the training of detectors by creating learning adversaries during training to explore the detector's weakness and update the detector's black-box features generated by Transformer for new attacks.
> > > - In CTA problems, there are combinatorially many cache access sequences that can constitute meaningful attacks. MACTA aims at training the detector to discover the invariant features of the attack sequence so that there are limited chances for *future* attackers to fully bypass the trained detector given a time limit, *even if a learning attacker has access to the detector's response oracle.* Section 5.2.3 in our submission demonstrates that the hardened detector is able to slow down the (exploiter) attacker’s learning and prevent high information leakage (attacks per episode) given attacker learning time limits.

---

> ### Author Response · Authors · 2022-11-11
> **Initial response to reviewer 2 (part II)**
>
> **Vague expression in the abstract**
>
> In our submission, “unseen” means “even during the learning phase of the agent” or “not specifically trained against”. While Apruzzese et al., (2020) claims a similar finding, their “unforeseen attackers” are generated by tweaking some parameters with human knowledge and the new samples lie *within* the training distribution (quotes the paper “alter the same features of E2 but with intermediate increments”). Such practice generates new adversarial samples in a more GAN-like way. Our evaluation considers samples generated by different algorithms (human heuristic and different learning algorithms) or random seeds, and these samples can be completely underexplored by one particular detector. We believe our finding of detecting “unforeseen” attackers still adds value to the ML community.
>
> **Reset secret**
>
> The real-world CTA attack targets guessing a multi-bit secret; after one bit of information is transmitted, the attacker will move to the next bit, which may be different from the previous bit. We assume the targeted secret is a randomly generated multi-bit string, so the next secret bit resets randomly after each guess. A similar practice is real attack trajectories such as prime+probe attack and the environment setting in AutoCAT(Luo et al., 2022).
>
> **Why did the authors consider such a simplistic cache setting?**
>
> Cache has a repetitive structure with multiple cache sets with an identical structure. With a detector agent working with the setting in our experiments, we can scale the detection by having parallel detectors each dealing with a portion of cache independently. Previous cache timing attack detection such as  Cyclone (Harris et al., 2019) divides the cache into buckets and monitors anomalous events in each of the buckets. There are strategies to scale the detection to larger caches if there is a detector on small caches.
>
> **Why is it that, in Table 3, the “IBR-PPO Detector” has 0 detection rate against ALL attacks (including the IBR-PPO attack)? Furthermore, it is unfair to “remove” the perfect detection rate of Cyclone from Table 3. Moreover, also concerning Cyclone, I do not agree with the statement that “fails to detect RL attackers”: its detection rate is 20%, which is only 5% less than the proposed MACTA; at the same time, Cyclone exhibits a much better false positive rate than MACTA (4.2 vs 11.3): in these cases, Cyclone is much better than what the authors describe.**
> - IBR-PPO detectors can not detect other attack policies except the one it is trained on (detection rate 98%+, the attacker is from the previous best response iteration).
> - We have optimized our code to tune down the false positive rate of MACTA detector to be at the similar level as cyclone and the results will be reflected in the next revision of the paper.
> - AutoCAT and IBR-PPO attackers are also RL attackers and Cyclone has 0% detection rate on all AutoCAT attackers, however, MACTA detectors can respond to most of the RL attackers. We will revise the related text to mitigate possible misunderstandings.
>
> **In Section 5.2.1, the paper states that “MACTA has low detection speed”: where is this reported? Note that I do not refer to “epochs” but to actual “time”. Furthermore, what the paper does not report (afaik) is the amount of computational resources required to develop (i.e., write code and train) MACTA: indeed, I am inclined to believe that implementing MACTA requires a huge effort – potentially superior than the one used to develop, e.g, Cyclone.**
>
> - “MACTA has low attack speed” is reflected in Table 1 as the number of attacks per episode is smaller than AutoCAT and IBR-PPO Attacker (both are RL attackers). Observing the MACTA attacker’s trajectories, it inserts more redundant steps to confuse the detectors so it can only complete attacks at a lower frequency/speed.
> - Although MACTA can take effort in coding and experimenting with the training, it is hard to directly compare our effort and Cyclone’s. In terms of manual efforts, both our detector and Cyclone require the interface between the hardware simulator and the ML model. Once the MACTA detector is trained, it can respond to a great variety of attacks without further training.
> - Cyclone requires great expert efforts and insights to derive and select the features used by the model, while MACTA can discover common patterns of the attackers from the raw memory access traces automatically.
>
> ---
> *References can be found in the submission or general response*

---

### Official Review · Reviewer_aHoC · 2022-10-24

**Confidence:** 2
**Correctness:** 4
**Technical Novelty And Significance:** 3
**Empirical Novelty And Significance:** Not applicable
**Recommendation:** 5

**Clarity, Quality, Novelty And Reproducibility:**

The paper is clearly written overall. On the novelty side, the paper is incremental from prior work. Since the code is not released yet, it's still unclear about the reproducibility.

**Strength And Weaknesses:**

Strength:

Interesting problem space to apply RL

Weaknesses:

Intellectual difference between AutoCAT [Luo et al., 2022] and MACTA is small. The paper overall is an extension of AutoCAT where detector is formulated as part of the RL problem. On this end, the formulation presented in this work is incremental to Luo et al.

The evaluation is thin. I like the neural architecture study. However, as a security focused paper, I look forward to more security analyses providing insights that why specific attack/defense succeeded or failed. Perhaps cases studies as in Luo et al. can help.


**Summary Of The Paper:**

This paper formulates cache timing channel attack and defense as Partial Observable Markov Games (POMGs). Evaluation shows that the attack/defense agents trained from POMG achieve better generalization against unseen detectors/attacks compared to other works. The authors also discover that applying Transformers as the neural encoder of policy nets performs significantly better than MLPs.

**Summary Of The Review:**

The paper definitely studies an important problem. However, overall I feel this paper does not contribute much on the intellectual side compared to prior work and the current evaluation is thin.

---

> ### Author Response · Authors · 2022-11-11
> **Initial response to reviewer 1**
>
> Thank you for reviewing our paper!
>
> **Comparison with AutoCAT**
>
> Please check the general response above, and we provide more details here.
> Our work studies a fundamentally different problem from AutoCAT (Multi-agent vs. Single agent), and AutoCAT is only designed for offensive purposes.
> - First, we study the multi-agent reinforcement learning problem, which differs substantially from the single-agent RL. Since multiple agents are also optimizing their policies, we must deal with non-trivial challenges like training stability and evaluation methodology. For example, a naive implementation of RL (IBR-PPO) can lead to cycles in policy learning, and the learned detector can not generalize.
> - The critical problem we were trying to address in the paper is, “Can we learn a detector that can detect unseen attackers and terminate malicious programs as soon as possible.” To tackle this problem, we novelly applied population-based multi-agent reinforcement learning to detectors and attackers with interactions with benign agents in the critical CTA security problem. The key intuition behind our method is that the attacker should exhaust the available policy space with the procedurally stronger detector, and both agents should learn to respond to all previous opponents during training.
> - Second, we contribute a multi-agent environment with more interactive agents (benign agents, heuristic attackers, heuristic detectors, and learnable detectors) than AutoCAT.
>
> **Evaluation of the trajectory**
>
> In general, the attackers are generating variants of the Prime+Probe attack. However, the MACTA attack sequences interleave the Prime phase and Probe phase in a random fashion, include more redundant memory accesses, and trigger random victim accesses. These patterns make the trajectory look more similar to benign programs to bypass the detector. We will provide a detailed trajectory analysis in our next revision of the paper soon! Please let us know if you think what can help the evaluations.

---

> ### Author Response · Authors · 2022-12-13
> **Trajectory Analysis in the Latest Revision**
>
> Dear reviewer, thanks again for providing valuable reviews for our paper! We would like to kindly remind you that, according to your constructive suggestions, we have provided trajectory analysis in our latest revision, where the MACTA attacker demonstrates more interesting disguise behaviors than AutoCAT or IBR-PPO attackers. The attacker trajectory visualization provides evidence that our detector is gradually getting stronger, which pushes the attacker to alter its behavior through the multi-agent training framework. Our framework has also evolved to be more robust during the rebuttal period. We sincerely hope you would consider increasing the score given the improved result and paper quality. Thank you very much!

---

### Author Response · Authors · 2022-11-11
**Responses to General Comments**

We thank the reviewers (R1, R2, R3) for their constructive comments! We will incorporate all the suggestions in our next revision of the paper. We are encouraged that the reviewers find MACTA to be an interesting application in general.

We address some common questions here:

**Novelty claim**

We acknowledge that the claim “the first to show a real-world hardware security problem can be addressed by jointly improving both attacks and detection” is a bit vague and can be misleading.

To be more precise, we are the first to introduce the hardware timing attack problem as a promising application of multi-agent RL (MARL), and show that MARL can be effectively applied to detect simulated CTA attacks with strong generalization.
- Compared to AutoCAT, which is a single-RL attacker, we optimize the attacker and the defender policies jointly, yielding a multi-agent formulation. Note that a successful application of multi-agent RL to this problem is a non-trivial task. Besides, we contribute a multi-agent environment with more interactive agents (benign agents, heuristic attackers, heuristic detectors, and learnable detectors) than AutoCAT.

- Apruzzese et al. (2020) (mentioned by R2) use single-agent RL to generate adversarial samples from base attack samples through feature perturbation and train the detector (classifier) with a GAN-style supervised formulation. In contrast, we truly model the attacker’s and the detector’s action sequences with Markov Decision Process and RL. In addition, compared to the generalization capability across different intermediate feature values shown in (Apruzzese et al., 2020), we show generalization w.r.t. unseen attackers/detectors that are previously proposed by researchers and are not coded in the MACTA’s training stage.
- A few closest general security works we listed in the submission either (1) apply MARL in fundamentally different scenarios where there are no benign participants (Eghtesad et al., 2020, Want et al., 2019), or (2) analyze game-theoretic strategies where state and action space are derived and overly simplified (Anwar et al., 2018).

**Lack of “Realistic” settings**

The reviewers expressed concern that limiting the study to two simultaneous programs was unrealistic. Although detectors need to pick up malicious entities from many programs in some settings,  in off-the-shelf hardware, one physical core can only allow at most two simultaneously running programs, and each L1/L2 cache is shared only by programs on a single core. Even more advanced processors have a limited number of programs executing simultaneously due to system scheduling. Moreover, well-established works such as CC-hunter (Chen & Venkataramani, 2014),  Cyclone (Harris et al., 2019), and PerSpectron (Mirbagher-Ajorpaz et al., 2020) have also done similar experiments distinguishing benign-benign program pairs and attacker-victim program pairs as well.

**Everything is simulated**

We follow many existing works on CTA detection that use a cache hardware simulator (CC-hunter(Chen & Venkataramani, 2014),  Cyclone (Harris et al., 2019), PerSpectron (Mirbagher-Ajorpaz et al., 2020) ), a common practice for studying hardware design before being fabricated. We provide key differences between a simulator and real hardware.
- In terms of cache logic behavior (including memory access and cache hit or miss), there is a negligible gap between a cache simulator and the real hardware.
- In terms of the actual latency, area, and power, there is a gap between a simulator and the real hardware. A detailed study on these aspects is beyond the scope of this work. However, evidence shows that generated attack sequences in the simulator can be applied to real hardware (Luo et al., 2022).
Note that it is non-trivial to put detectors on real hardware since it requires hardware modification that is impractical for off-the-shelf hardware. We leave this for future work.

**Other aspects**

Compared with prior work, our work applies recent advances in machine learning, including (Async) PPO, population-based MARL, and Transformer structure to a hardware security problem. We present the hardware-level timing-attack problem to the machine learning community with accessible tools that may inspire new findings in machine learning.

We are revising the paper to improve it based on the reviewers’ constructive criticism and will send updates as soon as the revision is complete.

------------
*Reference* (We list extra references here; other references can be found in the submission.)

Apruzzese, G., Andreolini, M., Marchetti, M., Venturi, A. and Colajanni, M., 2020. Deep reinforcement adversarial learning against botnet evasion attacks. IEEE Transactions on Network and Service Management, 17(4), pp.1975-1987.

---

### Author Response · Authors · 2022-11-19
**Revision Version 1 Summary**

Dear Reviewers,
    Thanks again for helping make the paper in a better shape! We provide a summary of updates with the revision here:
- Fixed several vague expression and novelty claims;
- Updated Tables and Figures with the tuned MACTA framework;
  - The main effort was to reduce the false positive rate of the MACTA detector.
  - Changed the episode length from 65 to 64, so we train the learning models and evaluate them again in the new episode length. The new statistics fall in a reasonable range around the previous submitted version;
  - Exploitability figures are updated due to change in the MACTA detectors;
- Added attackers’ and benign trajectories analysis in the appendix (section Appendix.3);
- Added attack evaluations on real hardware (section Appendix.4). The experiment was conducted under 1-set 8-way setting (different from the 8-set-1-way in the paper) due to the hardware resource and time constraint during the rebuttal period. We will add more comprehensive real-hardware results in the next revision of the paper;
- Moved the second part of section 2.1 to Appendix.1 “Why Cache Timing Attacks”.

---

### Author Response · Authors · 2022-12-13
**Updates on statistics [12/12/2022]**

Dear reviewers and ACs,

We would like to update you that we have made our training framework more robust and improved our results during the rebuttal period.
- We studied different latency representations in the RL detector's observation space, and optimized the latency encoding. We found a discrete latency encoding that can result in a lower variance in the RL detector’s generalizability across multiple training instances.
- In the previous version, the CC-Hunter detector demonstrates a high false alarm rate, which causes the method not to be directly comparable to other methods. We are able to tune its false alarm rate to be comparable to Cyclone and MACTA on training benign datasets, and we report the detection rates of CC-Hunter on different attackers as well as its false alarm rates on the test benign dataset.
- We checked all of our statements in the paper; while the new results are more robust and have better detection rate, the main observations remain unchanged.

Here are the updated statistics:

**Table 2: (1) The MACTA attackers’ number of attacks per episode is lowered due to stronger detectors.**

Table 2 Attacker Performance **(Updated, New setting)**
| Metric\Attacker | Prime+Probe    | AutoCAT | IBR-PPO Attacker | MACTA Attacker |
|---|:---:|:---:|:---:|:---:|
| Attacker Correct Rate | 100.0 $\pm$ 0.0 | 100.0 $\pm$ 0.1 | 99.7 $\pm$ 0.1    | 100.0 $\pm$ 0.1 |
| Attack per Episode    | 3.0 $\pm$ 0.0   | 5.2 $\pm$ 0.1   | 5.1 $\pm$ 0.1     | 3.4 $\pm$ 0.4 |

Table 2 Attacker Performance **(Paper, Old setting)**
| Metric\Attacker | Prime+Probe     | AutoCAT | IBR-PPO Attacker | MACTA Attacker |
|---|:---:|:---:|:---:|:---:|
| Attacker Correct Rate | 100.0 $\pm$ 0.0 | 100.0 $\pm$ 0.0 | 99.6 $\pm$ 0.3   | 99.8 $\pm$ 0.3 |
| Attack per Episode    | 3.0 $\pm$ 0.0   | 5.2 $\pm$ 0.1   | 4.9 $\pm$ 0.1    | 4.4 $\pm$ 0.3  |

**Table 3 and Table 4: (1) Higher and more stable detection rate of MACTA detector on all studied attackers than the previous result. (2) Lower CC-Hunter detection rate because we tune down its false alarm rate (3) MACTA attacker is more likely to be detected by MACTA detector and Cyclone because the attack policy/frequency is closer to Prime+Probe now. (4) MACTA attacker is less likely to be detected by the CC-Hunter as its signal doesn't contain the highly auto-correlated components that are detectable by CC-Hunter qualitatively.**

Table 3 Mean detection rate **(Updated, New setting)**
| Detector\Opponent | Prime+Probe    | AutoCAT  | IBR-PPO Attacker | MACTA Attacker  | Benign        |
|---|:---:|:---:|:---:|:---:|:---:|
| CC-Hunter         | 48.3 $\pm$ 0.6 | 20.6 $\pm$ 1.1 | 15.9 $\pm$ 1.8   | 11.2 $\pm$ 0.5  | 3.5 $\pm$ 1.9 |
| Cyclone           | -              | 0.0 $\pm$ 0.0  | 3.3 $\pm$ 2.2    | 33.3 $\pm$ 3.0  | 3.5 $\pm$ 0.9 |
| IBR-PPO Detector  | 0.0 $\pm$ 0.0  | 0.0 $\pm$ 0.0  | 0.0 $\pm$ 0.0    | 0.0 $\pm$ 0.0   | 0.8 $\pm$ 0.1 |
| MACTA Detector    | 97.5 $\pm$ 2.1 | 97.8 $\pm$ 2.4 | 98.2 $\pm$ 2.5   | 35.8 $\pm$ 18.0 | 3.4 $\pm$ 1.2 |

Table 3 Mean detection rate **(Paper, Old Setting)**
| Detector\Opponent | Prime+Probe     | AutoCAT  | IBR-PPO Attacker | MACTA Attacker  | Benign         |
|---|:---:|:---:|:---:|:---:|:---:|
| CC-Hunter         | 67.6 $\pm$ 1.3  | 50.1 $\pm$ 2.0  | 45.0 $\pm$ 1.6   | 43.4 $\pm$ 5.4  | 45.2 $\pm$ 4.6 |
| Cyclone           | -               | 0.0 $\pm$ 0.0   | 6.6 $\pm$ 3.1    | 12.0 $\pm$ 4.1  | 3.5 $\pm$ 0.6  |
| IBR-PPO Detector  | 0.0 $\pm$ 0.0   | 0.0 $\pm$ 0.0   | 0.0 $\pm$ 0.0    | 0.0 $\pm$ 0.0   | 0.8 $\pm$ 0.6  |
| MACTA Detector    | 81.2 $\pm$ 18.6 | 77.6 $\pm$ 20.4 | 71.7 $\pm$ 16.1  | 13.5 $\pm$ 14.7 | 2.1 $\pm$ 1.2  |

Table 4 Mean episode length **(Updated, New setting)**
| Detector\Opponent | Prime+Probe    | AutoCAT        | IBR-PPO Attacker | MACTA Attacker  | Benign         |
|---|:---:|:---:|:---:|:---:|:---:|
| IBR-PPO Detector  | 64.0 $\pm$ 0.0 | 64.0 $\pm$ 0.0 | 64.0 $\pm$ 0.0   | 64.0 $\pm$ 0.0  | 63.6 $\pm$ 0.1 |
| MACTA Detector    | 27.0 $\pm$ 2.1 | 15.2 $\pm$ 2.8 | 13.2 $\pm$ 4.2   | 47.0 $\pm$ 11.7 | 62.2 $\pm$ 0.6 |

Table 4 Mean episode length **(Paper, Old setting)**
| Detector\Opponent | Prime+Probe    | AutoCAT        | IBR-PPO Attacker | MACTA Attacker  | Benign         |
|---|:---:|:---:|:---:|:---:|:---:|
| IBR-PPO Detector  | 64.0 $\pm$ 0.0 | 64.0 $\pm$ 0.0 | 64.0 $\pm$ 0.0   | 64.0 $\pm$ 0.0  | 63.6 $\pm$ 0.1 |
| MACTA Detector    | 33.4 $\pm$ 18.4 | 33.7 $\pm$ 12.6 | 33.5 $\pm$ 11.8   | 58.4 $\pm$ 7.5 | 62.8 $\pm$ 0.7 |

**Figure 4: The general trend of Figure 4 is kept the same as the original Figure 4, and there is a constant gap between the number of attacks without a MACTA detector (5.2 attacks per episode) and with a MACTA detector (4 attacks per episode) if we train the exploiter for a very long time (1000 epochs). The difference is that it will take longer for an exploiter attacker to bypass the detector.**

Thank you very much for your great effort in reviewing our paper!

---

### Decision · Program_Chairs · 2023-01-20

**Decision:**

Accept: poster

**Justification For Why Not Higher Score:**

There were some weaknesses identified by the reviewers (see the above). Given the scores, this is the highest I could recommend.

**Justification For Why Not Lower Score:**

I think the paper makes some good contributions and experimental results are good. Authors did a good job in the rebuttals.

**Metareview: Summary, Strengths And Weaknesses:**

This paper formulates cache timing channel attack and defense as Partial Observable Markov Games (POMGs). Evaluation shows that the attack/defense agents trained from POMG achieve better generalization against unseen detectors/attacks compared to other works. The authors also discover that applying Transformers as the neural encoder of policy nets performs significantly better than MLPs.
Compared to ML approach, this work does not simply learn behaviors of possible CTA, but it allows attackers and detectors to interactively develop accordingly. However, there is lack of some theoretical explanations on MARL for CTA. Also one of the reviewers had some doubts that authors were able to simultaneously reduce the FPR while increasing the TPR (w.r.t. the previous version of the work). Authors should explain this more clearly and comprehensively in the final draft (aside from "we reduced the episode length").



**Note From Pc:**

if the above contains the word "oral" or "spotlight" please see: "oral" presentation means -> notable-top-5% and "spotlight" means -> notable-top-25%. As stated in our emails, we are disassociating presentation type from AC recommendations